

# Teaching climate risk: a pilot training for tertiary students and practitioners in Brazil

Pablo Borges de Amorim[1], Pedro Luiz Borges Chaffe[2]

[1]Graduate Program in Environmental Engineering, Federal University of Santa Catarina (UFSC), Florianópolis, 88.040-970, Brazil.
[2]Department of Sanitary and Environmental Engineering, Federal University of Santa Catarina (UFSC), Florianópolis, 88.040-970, Brazil.

*Correspondence to*: Pablo Borges de Amorim (borgesdeamorim.pablo@gmail.com)

**Abstract.** Climate change is one of the major challenges of our society thus educational resources on climate risk and adaptation are needed. In this case study, we present a short-duration training for tertiary students and practitioners about the Intergovernmental Panel on Climate Change (IPCC)'s climate risk framework. The training uses Problem-Based Learning (PBL) pedagogy, and its suitability and benefits are evaluated with observational qualitative analysis and self-assessment of

knowledge of the participants of 5 independent groups in Brazil. We find that the application of a mapping exercise using the IPCC's climate risk framework supports learning about climate risk, as well as data interpretation, creativity, teamwork, communication, and critical thinking by the participants. This work merges the IPCC's climate risk framework and PBL for climate risk training. The proposed training enables the teaching of climate risk in stand-alone courses and professional development training in areas which climate is an embedded component.

## 1. Introduction

Climate change is one of the major challenges that society is facing in the 21st century (Leal Filho, 2010; IPCC, 2018). The need for climate change education has been emphasized in several international agreements, such as the World Conference on Education for Sustainable Development in Aichi-Nagoya (Buckler et al., 2014), the United Nations Framework Convention on Climate Change, and the Paris Agreement (Leal Filho and Hemstock, 2019). To tackle the United Nations'

Sustainable Development Goal 13: Climate Action (SDG 13, United Nations, 2016), UNESCO (2017) suggests special attention should be given to climate risk management.

Climate risk management integrates climate-related information into decision-making to decrease the potential of damages and losses (Travis and Bates, 2014). National and international climate initiatives increasingly recommend climate risk assessments for adaptation planning (Brazil 2016, IPCC, 2018; Sherbinin et al., 2019; Travis and Bates, 2014). The climate

risk framework of the Intergovernmental Panel on Climate Change (IPCC) is commonly used to map risks and assess





adaptation options (IPCC, 2014; Sherbinin et al., 2019). However, these are complex activities that require not only factual knowledge about climate change but also problem-solving skills like the interpretation of data, teamwork, communication, creativity, and critical thinking (IPCC, 2014; Sherbinin et al., 2019; Travis and Bates, 2014). These skills can be improved with active learning (George et al., 2016; Lyon et al., 2013; McCright et al., 2013; Pierce, 2019; Pruneau et al., 2013).

Problem-Based Learning (PBL) is an instructional method wherewith students learn about a subject through the experience of solving a real problem (Wood 2003; Buckler *et al.*, 2014). The PBL is a student-centred pedagogy that does not focus on solving problems with a definite solution, but rather on developing desirable skills and attributes. It includes knowledge acquisition, interpretation of data, creativity, teamwork and communication (Prince, 2004; Buckler et al., 2014; Wood, 2003; Pruneau et al., 2013) which can be beneficial to climate change education (George et al., 2009; McCright et al., 2013; Pierce,

2019; Pruneau et al., 2013). A sound training and an effective educational courseware based on PBL can save time and resources and is likely to enhance the learning of tertiary students and practitioners (George et al., 2016).

The literature reveals the need for effective training and educational courseware on climate risk that stimulates problem-solving skills. Training to strengthen applied climate education and promote problem-solving skills are desirable in stand-alone courses and where climate is an embedded component (George *et al.*, 2016; McBean and Rodgers, 2010; Reid, 2019;

Yen *et al.*, 2019). In this paper, we describe the development and delivery of a short duration training based on the IPCC's climate risk framework and PBL. Further, we address the following research questions:

(RQ1) Is a training scheme based on PBL and mapping suitable to support students' learnings about the IPCC's climate risk framework?

(RQ2) What are observations and learnings from an introductory pilot training on climate risk which is targeting tertiary

students and practitioners?

We use Brazil as a case study because of the high demand for climate risk experts and the considerable number of courses and programs with the potential to include climate risk management (Cadastro Nacional de Cursos e Instituições de Educação Superior, 2020; Cursos da Pós-Graduação Stricto Sensu no Brasil [2017 a 2020], 2020). We test and evaluate the training scheme based on qualitative observational analysis and self-assessment of the participants' of five different groups

including tertiary students and practitioners.

## 2. Materials and methods

### 2.1 Training development

Given the need for enhancing learning about climate risk and adaptation for water professionals (George et al., 2006, 2016), the training was initially designed to supplement the existing Water Resources Planning discipline in the Sanitary and

Environmental Engineering bachelor course of the Federal University of Santa Catarina (UFSC). As new opportunities emerged, the training was delivered for graduate students of the Water Resources Management Master program at the State University of São Paulo (UNESP) and practitioners of the Secretary of Sustainable Economic Development of Santa





Catarina State (SDE). This provided an excellent opportunity for us to design effective training to enhance student's learnings about climate risk and adaptation.

The theoretical basis underpinning this training is PBL, which is a pedagogical method based on principles of adult learning theory and requires an emphasis on problem analysis, knowledge application, and cooperative work (Wood, 2003). Several studies find that PBL has a positive effect on learners' knowledge retention and interest in the subject (McCright et al. 2013), as well as on the development of problem-solving skills (Buckler et al., 2014; Prince, 2004; Pruneau et al., 2013; Wood, 2003).

The background information used in the training is the IPCC's climate risk framework. The IPCC is the most credible and comprehensive reference regarding climate change (Cooperation with the IPCC, 2021). The IPCC defines climate risk as the combination of *climate hazard*, with the *exposure* and *vulnerability* of a receptor (Figure 1), while *vulnerability* is comprised by *sensitivity* and *adaptive capacity* (IPCC, 2014). Maps representing climate, biophysical and socioeconomic data are a substantial part of the toolkit for assessing the risks of climate change and are frequently associated with the IPCC's

framework (Sherbinin et al., 2019). Mapping is used to identify geographic areas where impacts are expected to be the greatest and, consequently, may require adaptation innervations (Sherbinin et al., 2019). Mapping promotes the development of interpretation of data and teamwork skills (Sherbinin et al., 2019; Yen et al., 2019). At the same time, visualization in a form of a map improves understanding of how climate change interacts with society, raising awareness about the relevance of the issue (Preston et al., 2011). It complements adult learning theory where there is preference for problem-solving and

building on experiences (George et al., 2016).

## 2.2 Training delivery

The main goal of the training is to enhance learning about climate risk and adaptation and its connection to water planning. The intended learning outcomes are:

    (1)  Capacity to explain the IPCC's climate risk framework and differentiate its components;

(2)  Associate socioeconomic, biophysical, and climate data with the climate risk components;

    (3)  Map climate risk and classify risk zones;

    (4)  Assess adaptation options;

    (5)  Communicate via presentation the outcomes and the limitations regarding data sufficiency.

The learning outcomes are designed to improve competence in climate and water planning skills, i.e., interpretation of data,

creativity, teamwork, communication, and critical thinking, by constructing knowledge from experiences (Kolb, 2015). Educational material (available in the supplement) in the form of a courseware includes:

    •     a slide show presentation to introduce the IPCC's climate risk framework (Supplement 1)

    •     a set of maps for participants (Supplement 2)

    •     a script for facilitators, including cues to assist the delivery of the training sessions (Supplement 3)



The training lasts 120 minutes and is divided into five sessions (Figure 2), which are associated with the intended learning outcomes. The following subsections describe the training sessions in detail.

### 2.2.1 Session 1: Introduction to the IPCC's climate risk framework

The learning goal of this session is to introduce the IPCC and its climate risk framework. With the support of the slideshow presentation, the trainer introduces the IPCC and its climate risk framework, where risk is a combination of a *climate hazard*,

with the *exposure* and *vulnerability* of a system. *Vulnerability* is comprised of *sensitivity* and *adaptive* capacity (IPCC, 2014). The session, in the form of an expository lecture, is complemented by an interactive exercise to reinforce the learning outcome and demonstrate competency. A flood impact case is drawn on the blackboard and participants must answer questions about the components of risk (Figure 1Figure 3). The participants are asked to point out the most *exposed* element in the scene, the most *sensitive*, and the most *capable to adapt*. The session takes 20 minutes.

### 2.2.2 Session 2: Associate socioeconomic, biophysical, and climate data with the climate risk components

The learning goal of this session is to interpret socioeconomic, biophysical, and climate data in a form of maps and associate them to the IPCC's climate risk components. This and the following sessions are based on PBL pedagogy and apply the IPCC's climate risk framework to a fictional situation using mapping. The training simulates a situation where the federal government requests a consultancy service to: (i) identify areas in Brazil with significant climate risk, and (ii) assess

adaptation options to reduce the risk. Climate risk assessment for different sectors is a real demand of the Brazilian federal government, foreseen in the Brazilian National Plan for Climate Change Adaptation – NAP (Brazil, 2016). Since the focus is on water planning, it was considered the top four climate impact types according to the Brazilian NAP: floods, landslides, water scarcity, and agricultural droughts.

The PBL pedagogy recommends small working groups (Prince, 2004; Wood, 2003), thus, each working group has 3-6

members and addresses one type of impact. Each working group receives a set of 4 maps illustrating socioeconomic, biophysical, and climate data. Figure 4 shows the set of maps associated with water scarcity, which is an example of the database that working groups use to assess climate risks and adaptation options.

In this session, participants classify the maps into: (i) climate hazard, (ii) exposure, (iii) sensitivity, or (iv) adaptive capacity. It is important to ensure that the working groups correctly interpret the maps (e.g., theme, legend, and caption) and

understand how that data is associated with the IPCC's climate risk components. This session lasts 20 minutes.

### 2.2.3 Session 3: Map climate risk and classify risk zones

The learning goal of this session is to map climate risk and classify risk zones. Since climate risk is the combination of climate hazard, exposure, sensitivity, and adaptive capacity, participants estimate risk by overlapping the maps illustrating socioeconomic, biophysical, and climate data. Each working group receives a blank map of Brazil where they are asked to

illustrate and classify the risk zones (e.g., low, medium, or high risk). This session lasts 20 minutes.



### 2.2.4 Session 4: Assess adaptation options

The learning goal is to understand how risk can be reduced and thus assess adaptation options. The participants discuss the factors that most affect the level of risk and, based on them, propose actions to reduce the risk. Participants should focus on the data they received and reflect on the government's role in climate change adaptation (e.g., delimiting occupation areas,
implementing water resource planning, and reducing water distribution losses). This activity lasts approximately 20 minutes.

### 2.2.5 Session 5: Communicate via presentation

The learning goal is to communicate via presentation the outcomes of the previous sessions, as well as the limitations regarding data sufficiency. Group presentation is an essential part of the PBL process and is an opportunity to assess the participants' performance, in particular to the acquisition of abilities in problem-solving and professional competence
(Macdonald and Savin-Baden, 2004). The participants begin by presenting the classification of the maps and the reasons behind their choice. Next, they present the climate risk map and the criteria adopted for the definition of risk areas. In the end, the working group presents the adaptation options and discusses the data they received, including relevant missing data that could improve the analysis. Each working group has 5 minutes to present the results and ~ 3 minutes to answer questions from peers.

**2.2 Methods of Evaluation**

Numerous forms of assessment have been proven to be successful in PBL, ranging from group presentations, self-assessment, and written reports to non-traditional methods, such as tripartite assessment (Lyon and Teutschbein, 2011; Macdonald and Savin-Baden, 2004). Given the short duration of the training and the objective of this case study, which is to develop an effective and motivating training scheme, we use qualitative observational analysis (Katz-Buonincontro and
Anderson, 2018) and a simple self-assessment of knowledge of the participants (Andrade, 2019; Boud, 1995). The goal of the qualitative observational analysis is to assess the suitability of the training scheme and the educational courseware in supporting students' learning about climate risk and promoting the practice of problem-solving skills (RQ1). Annotations include the demonstration of knowledge acquisition and skills by the participants, as well as feedback and misunderstandings that may require courseware improvements (RQ2).

Self-assessment is the process by which the participants reflect on their learning (Andrade, 2019; Boud, 1995). It is an important aspect of learning and one of the most important skills for professional development (Boud, 1995; Lyon and Teutschbein, 2011; Macdonald and Savin-Baden, 2004). However, the results from self-assessments should be carefully interpreted. Participants tend to overestimate their competence, which may lead to unreliable estimations of knowledge acquisition (Andrade, 2019; Kruger and Dunning, 1999; Macdonald and Savin-Baden, 2004; Strobl et al., 2020). On the
other hand, self-assessment can increase the interest and motivation of participants for the subject and support them in developing critical skills for analysis of their work and performance (Andrade, 2019; Boud, 1995), which are aligned with





the PBL principles (Macdonald and Savin-Baden, 2004). In this case study, the self-assessment comprises a simple survey distributed before and after the training sessions where participants had to rate their level of knowledge about climate risk on a scale ranging from 'nothing' to 'very much'. Assessing the participants' perception of knowledge acquisition supports

answering the RQ1.

## 3. Results

### 3.1 Training development

The training courseware was developed along with its delivery to 5 independent groups in Brazil from 2018 to 2019 and reached 94 higher education students and practitioners. Three groups are composed of undergraduate students of the Water

Resources Planning discipline of the Environmental and Sanitary Engineering bachelor course at UFSC. One group is of graduate students of the Water Resources Management Master program at the UNESP. The last group is of practitioners of the SDE. Figure 5 shows the timeline of training and the background information, i.e., the institution, the number and the profile of participants, the duration of the training, the training modules, and the documentation available. In some trainings, the content can be extended if time is available. In the third training, for example, introductory lectures on climate change

and climate information were added. These include climate scenarios in the climate risk mapping session as well. In the fifth training, an interactive lecture to introduce the scientific basis of climate change was added.

### 3.2 Training delivery

#### 3.2.1 Session 1: Introduction to the IPCC's climate risk framework

In general, it was observed that the participants could answer the questions correctly, suggesting that the expository lecture

complemented by an interactive exercise was useful for introducing the IPCC's climate risk framework and its elements.

#### 3.2.2 Session 2: Associate socioeconomic, biophysical, and climate data with the climate risk components

The participants were familiar with maps illustrating socioeconomic and biophysical data, but they had difficulties interpreting some climatology maps. The participants' queries were mainly about climate indices, such as the maximum number of consecutive dry days. Overall, the participants were able to classify the maps satisfactorily, especially for

*adaptive capacity*, demonstrating the students' gain of knowledge about the IPCC's climate risk framework (RQ1). It was noticed that students practiced the interpretation of data (RQ2). Difficulties emerged when classifying *exposure*, which is the most difficult aspect to be classified while mapping (Sherbinin et al., 2019). *Exposure* is the geographic location of a receptor, or a system, in relation to the *climate hazard* (IPCC, 2014). In mapping, to estimate *exposure* it is necessary to combine the spatial distribution of the receptor with the spatial distribution of the *climate hazard* (Sherbinin et al., 2019).



### 3.2.3 Session 3: Map climate risk and classify risk zones

The working groups proposed several types of climate risk maps (Figure 6). In the first training, the risk maps (Figure 6a, d) were much simpler than in later applications (Figure 6b, c, e, f). From the second training onwards, the participants were explicitly informed about the possibility of using a scale of risk (e.g., low, medium, and high) resulting in more sophisticated outcomes in the second and third trainings than those of the first group. In the second and third training, the working groups in charge of landslides were able to differentiate the levels of risk, with a detailed risk zoning (Figure 6b and 6c). For agricultural droughts, the working groups included a weighting scheme, similar to a multi-criteria analysis (Figure 6e and 6f). The complete lists of outcomes from all five trainings, as well as the maps provided as input for the mapping activity, are available in the Appendix A. All working groups were able to map climate risk zones, confirming the usefulness of the educational courseware for enhancing learning about the IPCC's climate risk framework (RQ1). The classroom observations suggest that this session promotes teamwork and creativity (RQ2). There were intense discussions between the participants and decisions were based on the working group consensus. At the same time, the outcomes of the working groups were very diverse (e.g. Figure 6), which is an indicator of creativity (Katz-Buonincontro and Anderson, 2018).

### 3.2.4 Session 4: Assess adaptation options

In many cases, the proposed adaptation options were satisfactory. For example, in the first training, the working group responsible for the flood case proposed increasing *adaptive capacity* by enhancing education as a means to reduce illiteracy (a map provided as input). For landslides, all working groups suggested reducing *exposure* by zoning areas of non-human occupation. In some cases, the working groups presented adaptation options based on their personal beliefs rather than on the evidence provided by the data given (i.e., the set of maps). The results suggest this session enhances knowledge about the usefulness of the IPCC's climate risk framework for climate change adaption (RQ2). At the same time, it promotes the practice of critical thinking (RQ2), by identifying the limitations of their work and proposing additional data needed.

### 3.2.5 Session 5: Communicate via presentation

The group presentations were satisfactory and demonstrated that the participants understood the principles of the IPCC's climate risk framework (RQ1). After instructions towards a more focused presentation (following the order of the sessions), the participants were able to communicate their results in a clear and consistent fashion (RQ2). Presentations were generally made by more than one working group member (Figure 7), a decision that demonstrates teamwork (RQ2).

### 3.2.6 Participants' perception of knowledge acquisition

To estimate the effect of the training on the participants' perception of knowledge change, a simple self-assessment pre- and post-training sessions was applied. Figure 8 illustrates two examples of the students' self-rating. In both trainings, participants self-rated scores pre- and post-training rose from "nothing" to "medium" or "a lot". The self-assessment results



demonstrate that the proposed training scheme, based on PBL and mapping, was adequate to support participants' learnings about the IPCC's climate risk framework (RQ1). It was clear that the participants perceived a significant level of knowledge gain (Figure 8) and the training produced a considerable level of satisfaction and motivation, an expected outcome from active learning (George et al., 2016; Lyon et al., 2013; Prince, 2004; Valaitis et al., 2005; Wood, 2003). Indeed, the practical sessions, the teamwork environment and the self-assessment were positively recognized. One student of the fourth training

said: "*I really liked the practical part. I think the concepts were very clear to me after that and it was really nice to discuss in groups those maps.*". In the third training, one student said: "*Thanks for the training. It was the first time I had the opportunity to work with my classmates.*". In the second training, one student said: "*Very didactic, and I really liked that activity of marking our level of knowledge about the subject before and after the activity.*". Despite the gain of knowledge perceived by the participants and the positive feedbacks, self-assessments tend to overestimate the competence of the

participants (Andrade, 2019; Boud, 1995; Kruger and Dunning, 1999; Strobl et al., 2020). However, self-assessment promotes personal development, which is interlinked with the acquisition of content (Andrade, 2019; Boud, 1995). Its application is advantageous for learners and trainers, such as feedback, student engagement, and increased trust (Taras, 2010), as observed in the delivery of this training (RQ2).

**4. Discussion**

The most significant findings of this research are:

    (1) Training based on the PBL and mapping can be used to support students' learnings about the IPCC's climate risk framework. The training produced high levels of satisfaction and the participants perceived a gain in knowledge (RQ1)

    (2) Training based on the PBL, mapping and the IPCC's climate risk framework fosters the practice of interpretation of

data, creativity, teamwork, communication, and critical thinking (RQ2)

    (3) The delivery of the training to five different groups provided several lessons learned and, consequently, improved the training and the educational courseware (RQ2).

This will now be discussed further concerning pedagogical skills, the benefits of using PBL in climate risk training, and limitations of the training.

**4.1 Pedagogical skills**

Active learning requires a profound and comprehensive factual knowledge about the target topic (George et al., 2009; Prince, 2004). In this case study, both trainers had more than ten years of experience in climate change and water resources planning, which may have contributed to the successful delivery of the training. George *et al.* (2009) recommend that trainers have a long experience in climate sciences. They highlight the lack of technical competence on climate risk by





trainers and, to overcome this, they suggest focusing on general problem-solving skills and knowledge about key concepts relating to climate risk.

Three major pedagogical skills necessary for the delivery of this training are discussed, which are: facilitation, ability to stimulate students, and preparation of educational courseware.

First, the facilitation script (Supplement 3) was useful in conducting the sessions successfully. Active learning is self-
directed learning on the part of the students and facilitation is essential (George et al., 2009; Wood, 2003). That means the educator needs abilities in reflective dialogue with students, in guiding students to identify connections and bring balance to discussions (May, 2000).

Second, the working groups were limited to a maximum of six members, and clear and short instructions were provided in each session to ensure engagement by the participants. Active learning requires extra attention to prepare activities that
stimulate participation and engagement (George et al., 2009; Prince, 2004; Wood, 2003). Several authors (Prince, 2004; Weber et al., 2021; Wood, 2003) highlight the importance of working in small groups and providing clear instructions to guarantee participation and engagement.

Third, the preparation of the maps (search, selection, and adjustment) was the most time-consuming activity. The set of maps was updated as the need for improvements was perceived. Indeed, PBL requires a significant amount of resources (e.g.,
printed materials) and time for preparation (George et al., 2009; Prince, 2004; Wood, 2003).

In addition, trainers should consider specific pedagogical practices for online trainings (Bailey and Card, 2009). A challenge that arises is how to apply active learning in a circumstance that demands social distancing, for instance, the 2020's COVID-19 educational disruption (Farnell et al., 2021). Like Orrill (2002) and Valaitis *et al.* (2005), the use of digital workspaces for visual collaboration should be explored in future works.

**4.2 PBL for teaching climate risk**

In this case study, the PBL approach nurtured the practice of interpretation of data, creativity, teamwork, communication, and critical thinking. Assessing climate risks is a complex activity that requires comprehensive interpretation of the context and the data available, as well as collaboration between different knowledge groups and creativity to communicate the results to multiple stakeholders (George et al., 2016; IPCC, 2014; Sherbinin et al., 2019; Travis and Bates, 2014). Many
education professionals consider active learning an effective learning approach that brings multiple side benefits (Buckler et al., 2014; Wood, 2003). The benefits of it in teaching climate risk are recognized by several authors (George et al., 2009; McCright et al., 2013; Pierce, 2019; Pruneau et al., 2013). Active learning also promotes enthusiasm in the classroom and high levels of satisfaction among students (George et al., 2016; Lyon et al., 2013; Prince, 2004; Wood, 2003), which were observed in the delivery of this training.





### 4.3 Limitations of the training

This training is a work in progress and will be further developed. The limitations of the current training with regards to climate risk education are discussed here to identify possible paths forward.

Risk mapping using the IPCC's climate risk framework is usually adequate for spatial planning (Sherbinin et al., 2019). Other frameworks are available and, in many cases (e.g., Damania *et al.*, 2010; George *et al.*, 2006, 2016, 2019; Mira-Salama *et al.*, 2013), the climate change risk matrix framework and the international guidelines for risk management (i.e., ISO 31000, 2018) are more meaningful. George *et al.* (2016) found that the climate change risk matrix is a suitable approach for enhancing water professionals' learnings about climate risk. It is important to note that the training presented here focused on water planning and was limited to 120 minutes, which was adequate to introduce the principles of the IPCC's climate risk framework. However, in further extensions of this training it is recommended to include the teaching of the climate change risk matrix and international guidelines for risk management, similar to George *et al.* (2016).

The training does not provide background information about climate change principles. George *et al.* (2016) emphasized that participants must be aware of the current state of knowledge on climate change. It is recommended a lecture to inform that human-induced climate change is supported by multiple sources of scientific evidence and related damages and losses can no longer be ignored (IPCC, 2018). Such materials were delivered in the third and fifth training and were very useful to raise awareness about the need for climate action.

The training does not address climate change scenarios. Although recommended to enable decision-makers to plan for near- and long-term time horizons (IPCC, 2014)*,* including climate change scenarios into climate risk mapping is not trivial (Sherbinin et al. 2019). The main challenges regard the interpretation of climate change projections and associated uncertainties (Sherbinin et al., 2019; Sutton, 2019). The third training included a session to map climate change risk scenarios, where a package of information with four different sources of climate change evidence was provided (i.e., trends in observational data, level of agreement on the signal of change from a climate model ensemble, regional projections, and literature synthesis). This activity required a considerable level of knowledge about climate change projections and associated uncertainties, and some difficulties in accomplishing the activity persisted even after an additional lecture about it. In a professional development climate course, George *et al.* (2009) reported that even trainers have difficulties in understanding climate change projections. Perhaps, an alternative is to reduce the package of information. George *et al.* (2016) simplified complex climatological data and obtained satisfactory results.

In terms of the methods of evaluation, it should be noted that in this case study we have not applied semi-structured qualitative interviews or statistical methods of analysis. This limits to some extent the ability to isolate the effect of the training proposed (Lyon et al., 2013). Triangulation of results from different evaluation methods enhances the validity of the conclusions (Grove et al., 2013; Lindorff and Sammons, 2018) and is recommended in future deliveries of this training. While this is a shortcoming of this case study, it highlights the value of including pedagogical experts in the design of lecture evaluations (Lyon et al., 2013), for instance in further developments of this training.



## 5. Conclusions

This case study demonstrates the development and delivery of a climate risk training based on PBL pedagogy where the IPCC's climate risk framework is applied through mapping. The training supports student's learnings about climate risk even in a short duration course. The training was delivered to 5 independent groups in Brazil from 2018 to 2019 and reached 94 higher education students and practitioners, which permitted gathering several lessons learned and, consequently, improving the training and the educational courseware. The self-assessment and the observational qualitative analysis show that the participants perceived a gain in knowledge and practiced interpretation of data, creativity, teamwork, communication, and critical thinking skills. At the same time, the results suggest that the practical sessions produced significant levels of satisfaction and motivation.

The educational courseware can be directly transferred to higher educational institutions in Brazil and will require minor adjustments when applied in other countries (e.g., updating of the maps). Regarding the pedagogical skills needed to deliver this training, it is recommended some level factual knowledge about climate change and water resources planning as well as facilitation skills and time for preparing the courseware. We recommend the evaluation methods adopted in this case study to be further developed for future application and evaluations of this training. Further research should investigate the value of a longitudinal study to assess what has been learned and applied and how learning can therefore be enhanced (e.g., up-scaled and accelerated). Through more targeted educational courseware in development, delivery, and evaluation in programmatic and integrative ways, climate risk and adaptation can be strengthened in stand-alone courses and professional development training where climate is an embedded component. This training, aligned with other structured courses, can be of value to educational systems and across cultures, and have appeal primarily to personnel in academia and curriculum designers as an indicator of applied climate knowledge and skills.

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

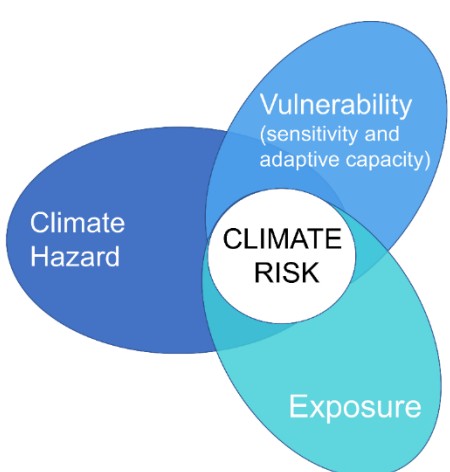

**Figure 1. IPCC's conceptual framework of climate risk (adapted from IPCC, 2014).**





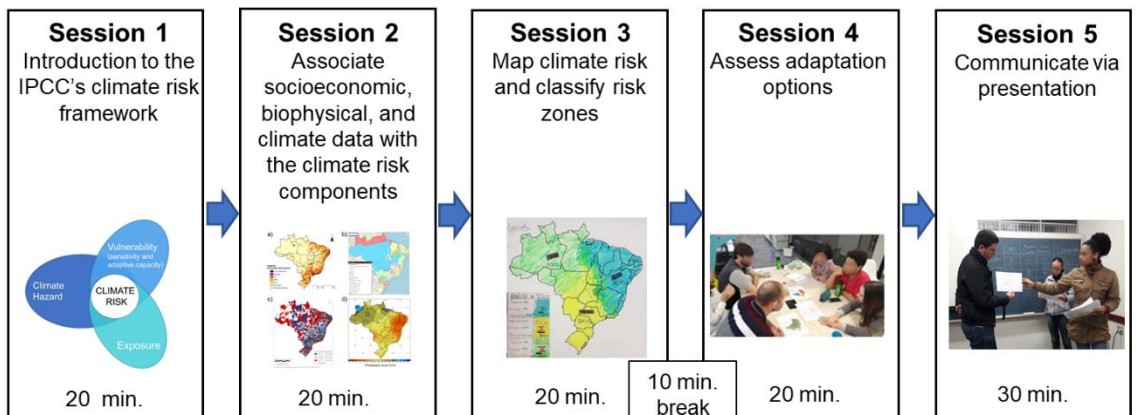


**Figure 2. Scheme of the training sessions (adapted from IPCC, 2014; MMA, 2017; Planos de Recursos Hídricos, 2020; Brasil, 2017; Normais climatológicas do Brasil, 2020)**

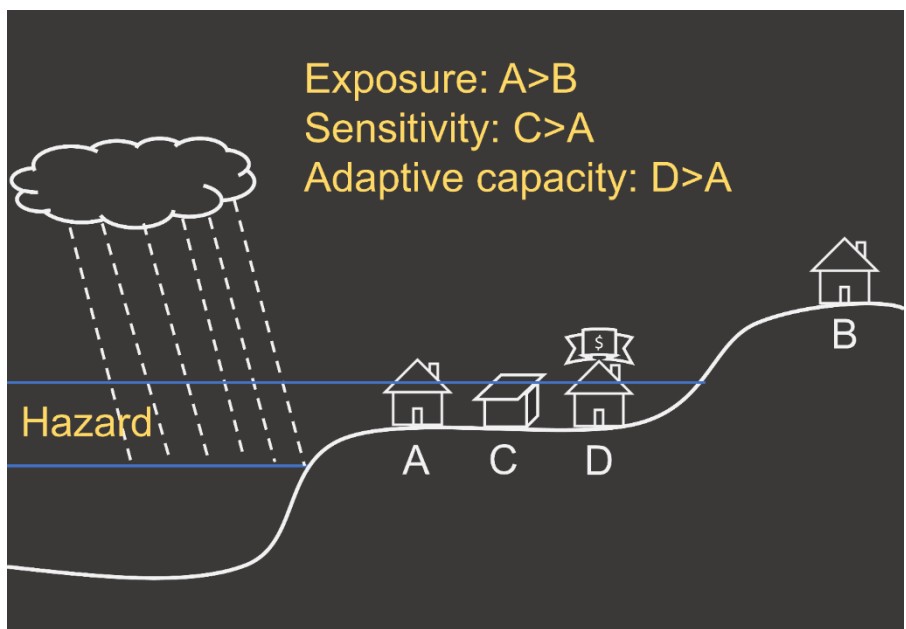

**Figure 3. Example of a classroom blackboard used to illustrate the climate risk exercise. Given a flood situation, house A is more exposed than house B; the wooden house C is more sensitive than the brick house A; the house with insurance D has a higher adaptive capacity than a house without insurance. The blue line indicates different flood levels.**



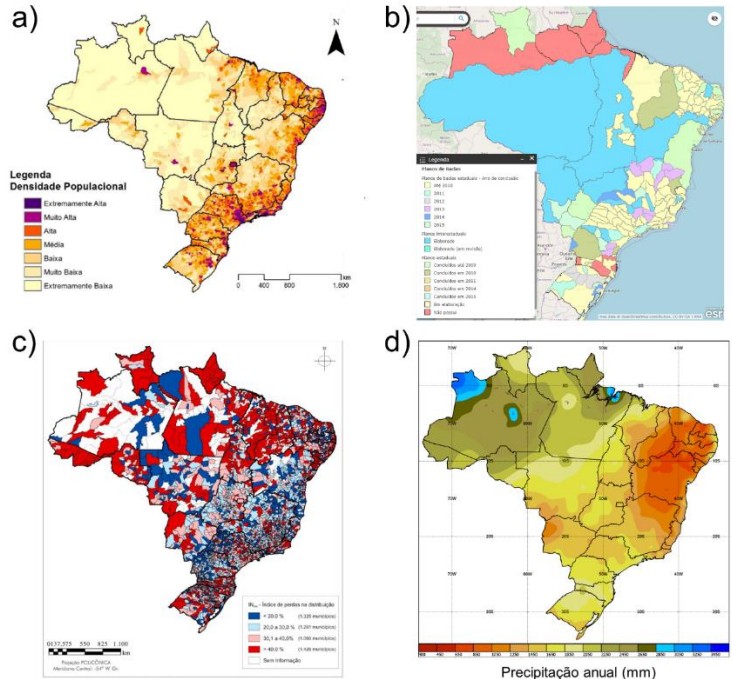

**Figure 4. Example of a map collection used for mapping climate risk associated with water scarcity. The maps illustrate: (a) population density (MMA, 2017); (b) stage of implementation of the water resources planning (Planos de Recursos Hídricos, 2020), (c) water distribution loss index (Brasil, 2017); (d) annual rainfall (Normais climatológicas do Brasil, 2020)**



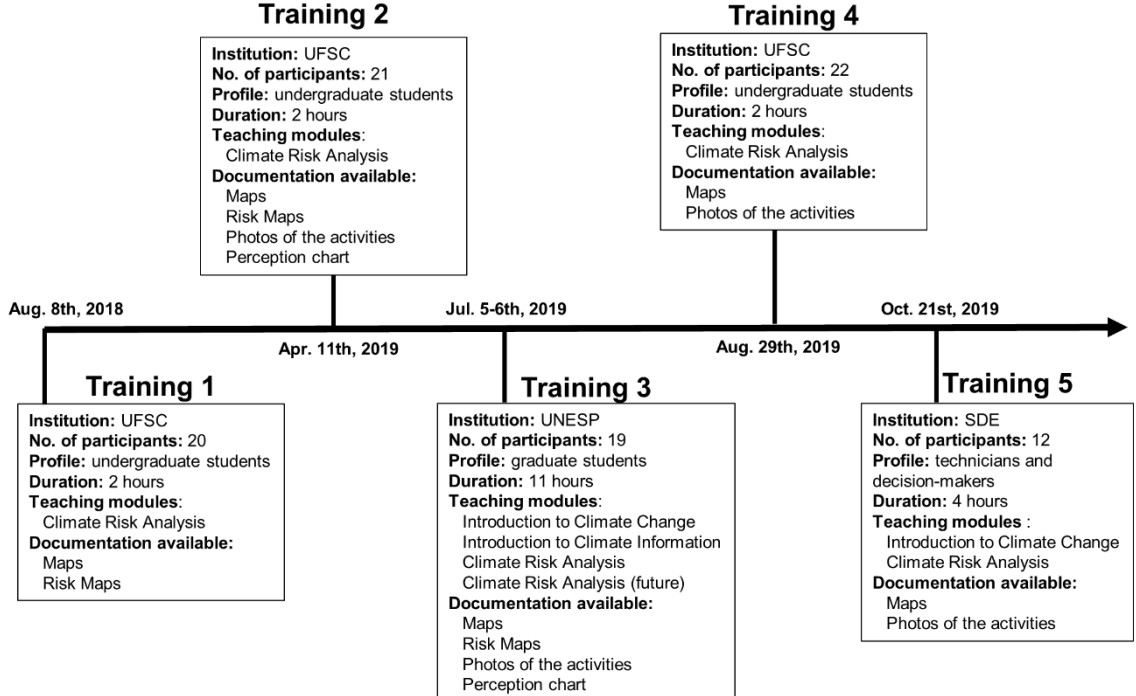

**Figure 5. Description of the 5 trainings delivered along 2018 and 2019. Training 1, 2, and 4 addressed undergraduate students from UFSC; training 3 comprised graduate students from UNESP; training 5 was made up of practitioners from the SDE.**



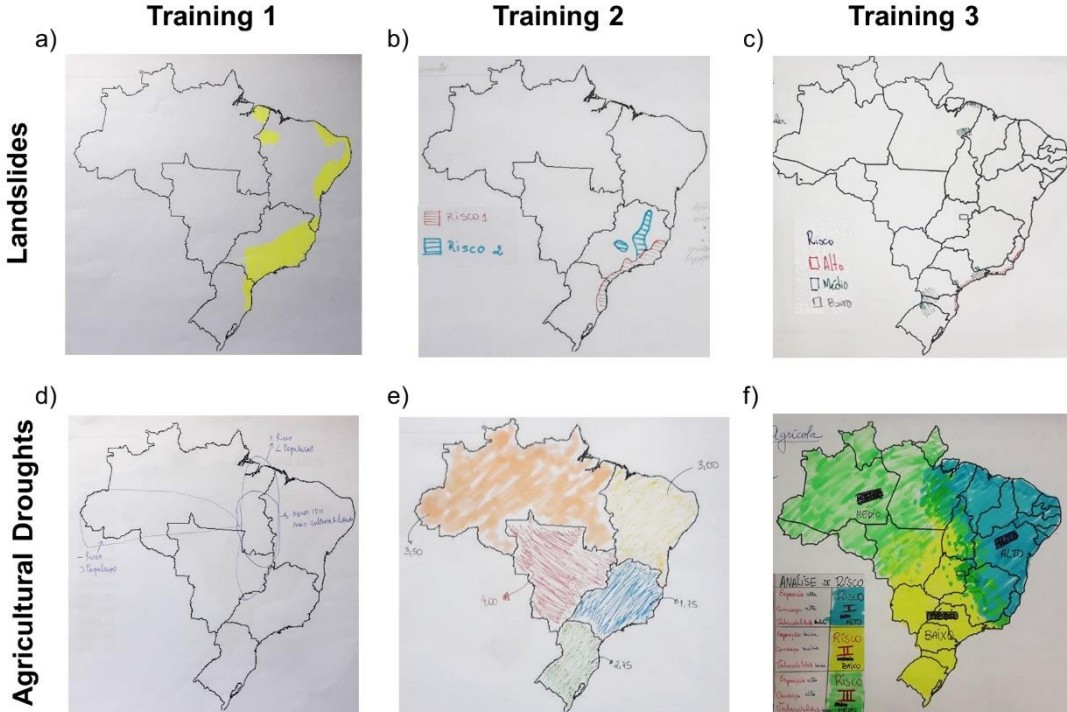

**Figure 6. Outcomes from the climate risk mapping for landslides and agricultural droughts working groups (rows) from 3 trainings (columns).**



a) **Training 2 | Water Scarcity**

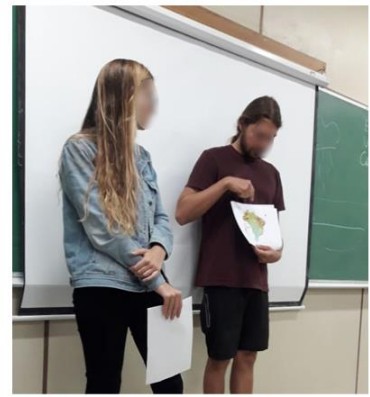

b) **Training 3 | Agriculture Droughts**

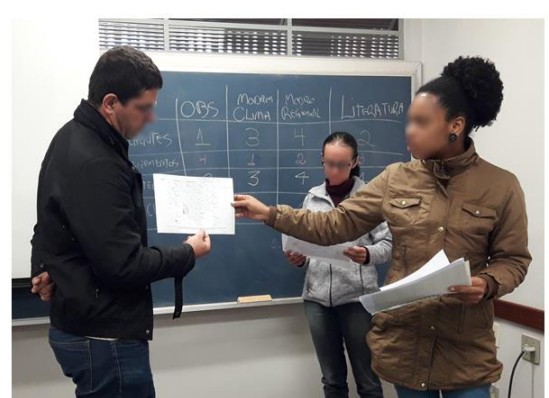

c) **Training 4 | Landslides**

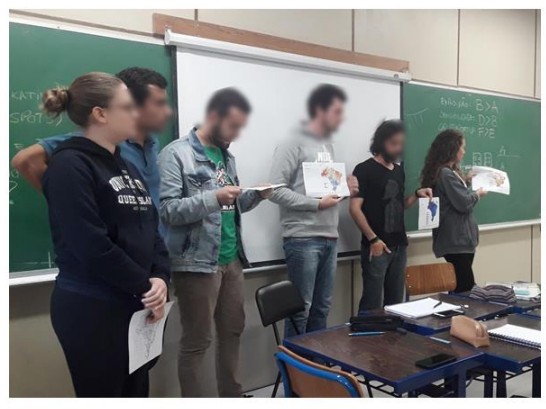

d) **Training 5 | Agricultural Droughts**

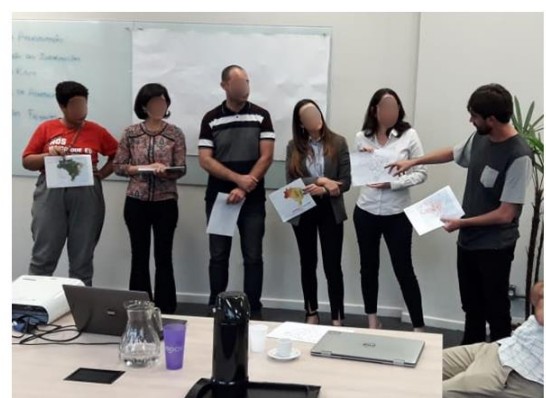

**Figure 7. Photos of the participants presenting their results.**



a)

**Training 2**

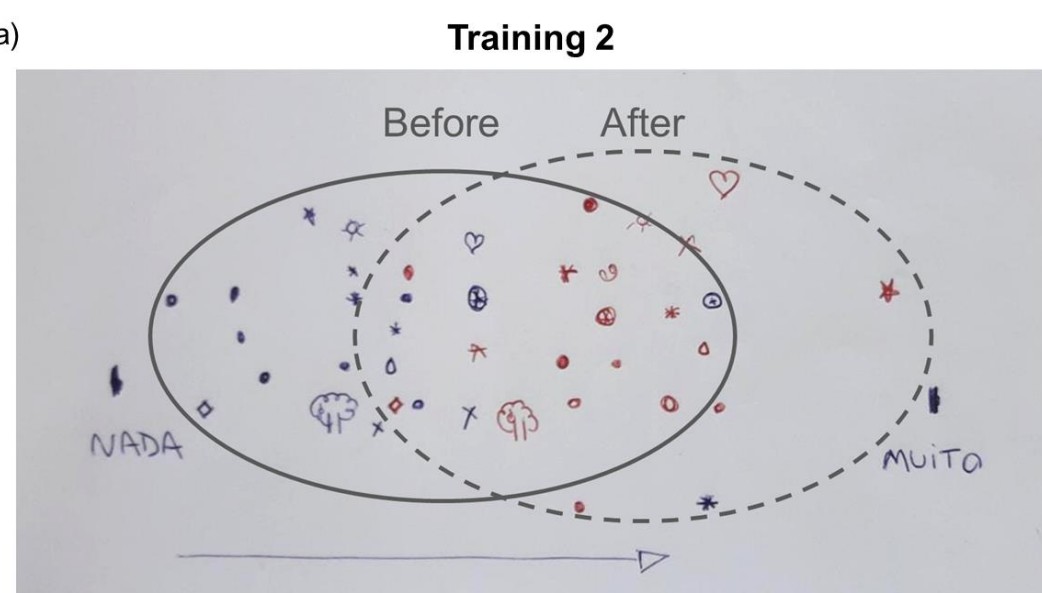

b)

**Training 3**

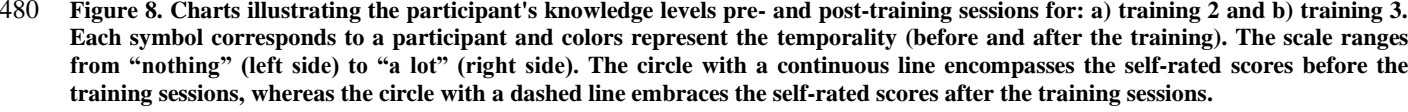

**Figure 8. Charts illustrating the participant's knowledge levels pre- and post-training sessions for: a) training 2 and b) training 3. Each symbol corresponds to a participant and colors represent the temporality (before and after the training). The scale ranges from "nothing" (left side) to "a lot" (right side). The circle with a continuous line encompasses the self-rated scores before the training sessions, whereas the circle with a dashed line embraces the self-rated scores after the training sessions.**





# Appendix A: lists of outcomes from all five trainings

**Training 1**

Date: August 8th, 2018

Institution: Federal University of Santa Catarina (UFSC)

Number of participants: 20

Profile: undergraduate students from Sanitary and Environmental Engineering

Duration: 2 hours

Teaching modules: Climate Risk Analysis

Documentation available: maps and risk maps

## Training 1 | Working Group Floods

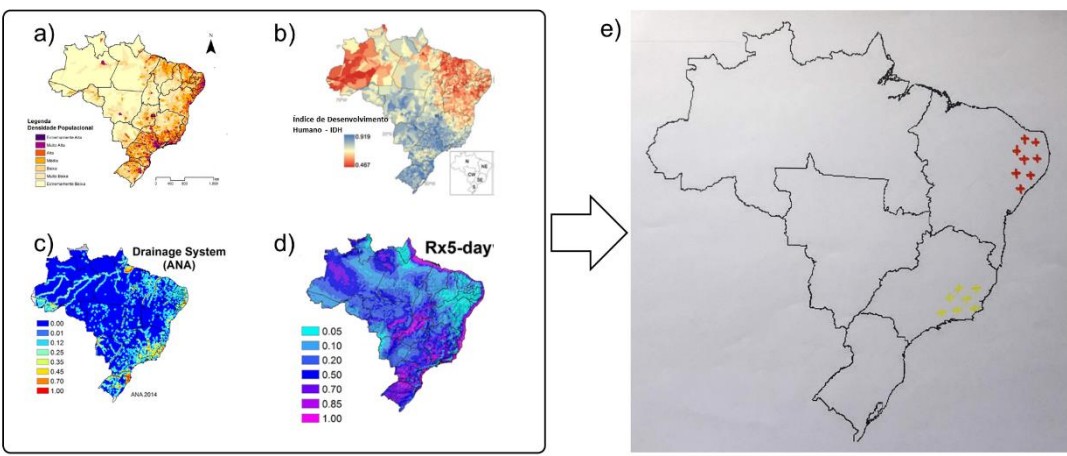

**Figure B1. Package of maps used in the first training for mapping climate risk associated with floods. The maps illustrate: a) population density (MMA, 2017); b) Human Development Index (HDI, Torres et al., 2012), c) watershed drainage (Debortoli et al., 2017); d) maximum consecutive 5-day precipitation (Rx5-day, Debortoli et al., 2017); and e) the climate risk map drawn by the participants.**





## Training 1 | Working Group Landslides

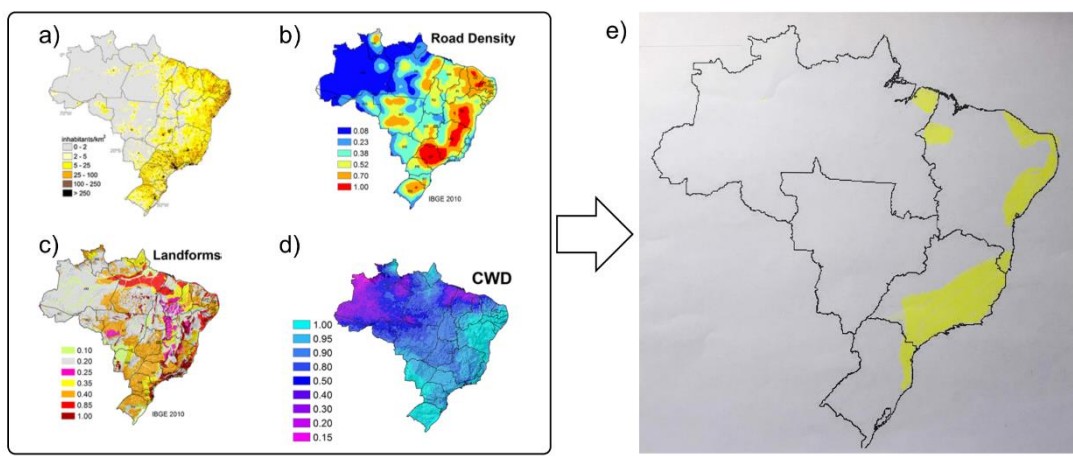


**Figure B2. Package of maps used in the first training for mapping climate risk associated with landslides. The maps illustrate: a) Brazilian population density (Torres et al., 2012); b) road density (Debortoli et al., 2017); c) landforms (Debortoli et al., 2017); d) Consecutive Wet Days (CWD, Debortoli et al., 2017); and e) the climate risk map drawn by the participants.**

## Training 1 | Working Group Water Scarcity

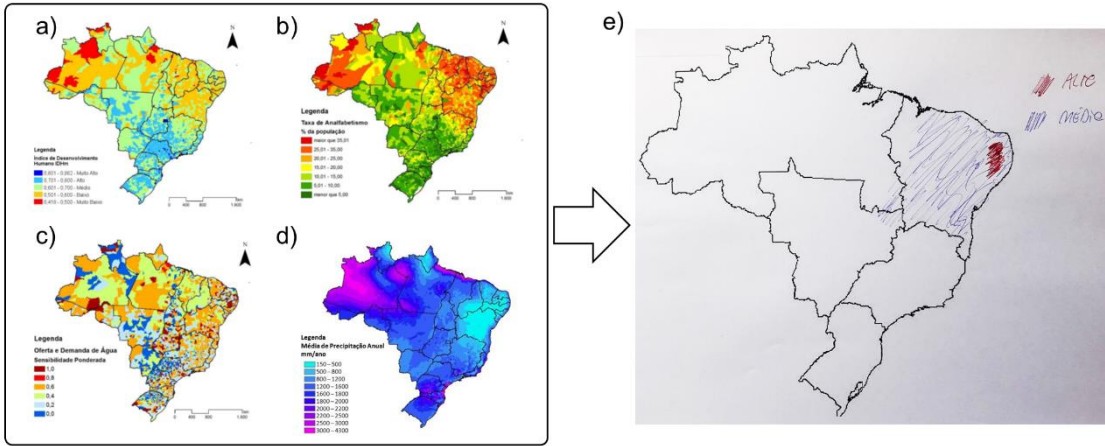


**Figure B3. Package of maps used in the first training for mapping climate risk associated with water scarcity. The maps illustrate: a) Human Development Index (HDI, MMA, 2017); b) illiteracy rate (MMA, 2017); c) water supply and demand (MMA, 2017); d) annual precipitation (MMA, 2017); and e) the climate risk map drawn by the participants.**






## Training 1 | Working Group Agricultural Droughts

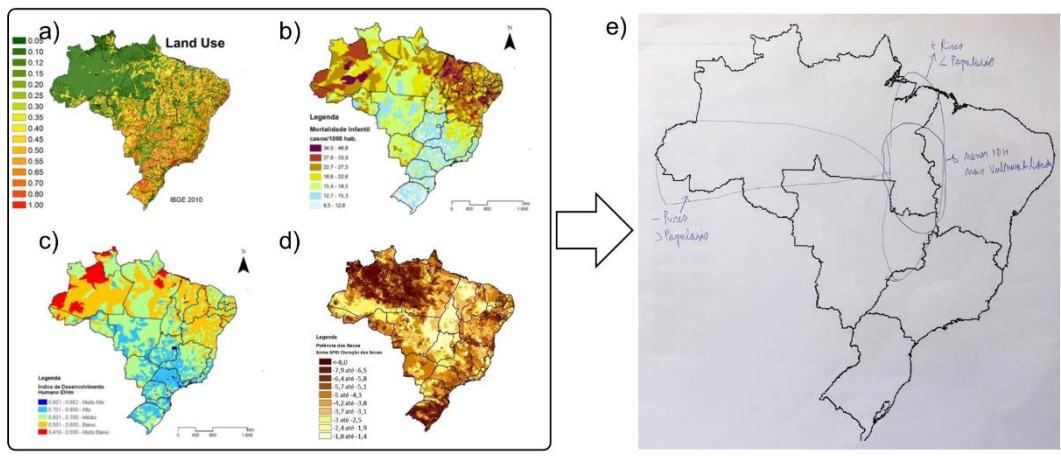

**Figure B4. Package of maps used in the first training for mapping climate risk associated with agricultural droughts. The maps illustrate: a) land use (MMA, 2017); b) child mortality (MMA, 2017); c) Human Development Index (HDI, MMA, 2017); d) drought potential (MMA, 2017); and e) the climate risk map drawn by the participants.**


**Training 2**

Date: April 11th, 2019

Institution: Federal University of Santa Catarina (UFSC)

Number of participants: 21

Profile: undergraduate students from Sanitary and Environmental Engineering

Duration: 2 hours

Teaching modules: Climate Risk Analysis

Documentation available: maps, risk maps, photos of the activities, perception chart



## Training 2 | Working Group Floods

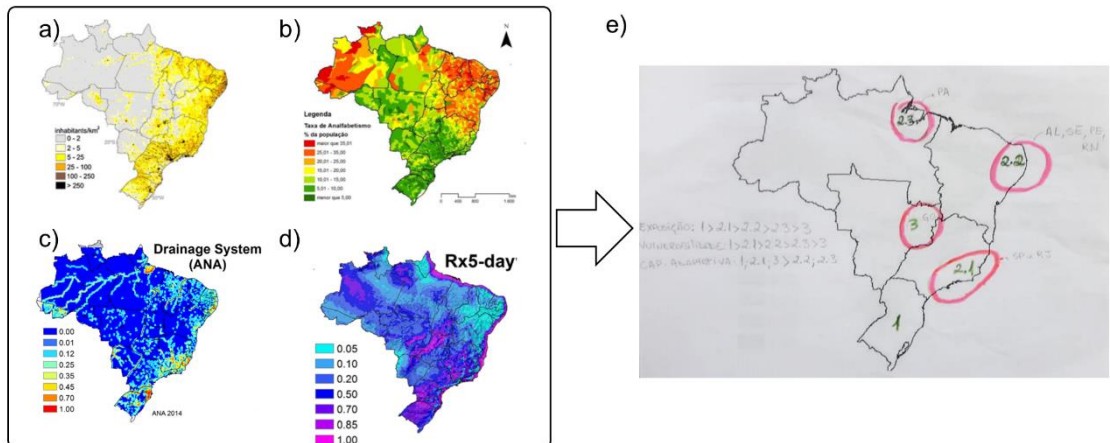

**Figure B5. Package of maps used in the second training for mapping climate risk associated with floods. The maps illustrate: a) Brazilian population density (Torres et al. 2012); b) illiteracy rate (MMA 2017); c) watershed drainage (Debortoli et al. 2017); d) maximum consecutive 5-day precipitation (Rx5-day, Debortoli et al. 2017); and e) the climate risk map drawn by the participants.**

## Training 2 | Working Group Landslides

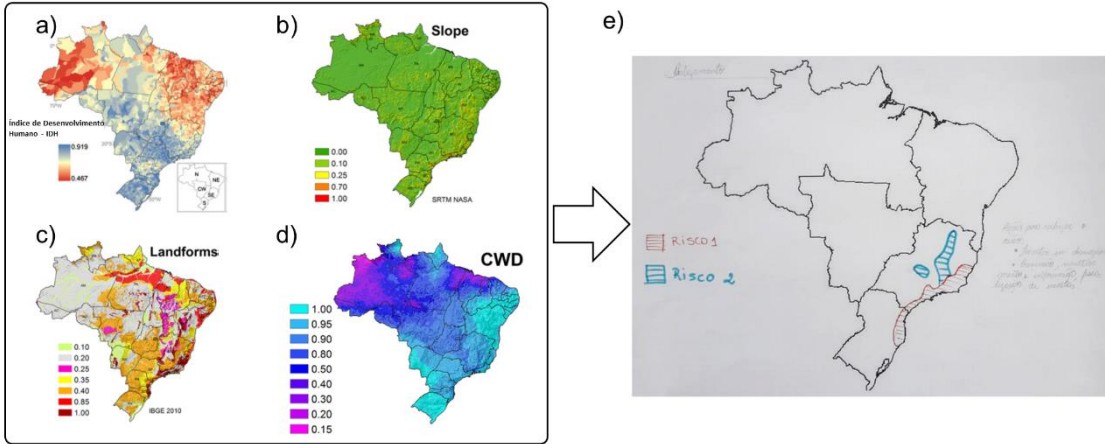

**Figure B6. Package of maps used in the second training for mapping climate risk associated with landslides. The maps illustrate: a) Human Development Index (HDI, Torres et al. 2012); b) slope (Debortoli et al. 2017); c) landforms (Debortoli et al. 2017); d) Consecutive Wet Days (CWD, Debortoli et al. 2017); and e) the climate risk map drawn by the participants.**





## Training 2 | Working Group Water Scarcity

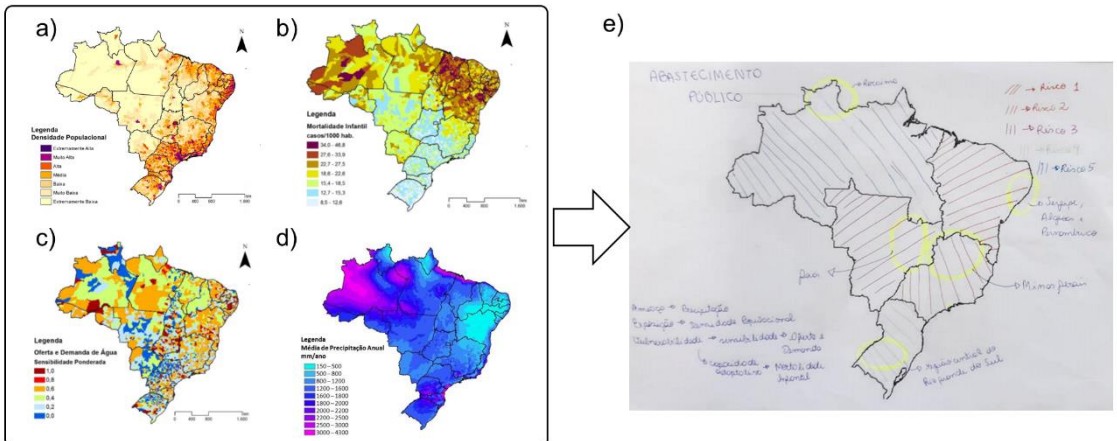


**Figure B7. Package of maps used in the second training for mapping climate risk associated with water scarcity. The maps illustrate: a) population density (MMA, 2017); b) child mortality (MMA 2017); c) water supply and demand (MMA 2017); d) annual precipitation (MMA 2017); and e) the climate risk map drawn by the participants.**

## Training 2 | Working Group Agricultural Droughts

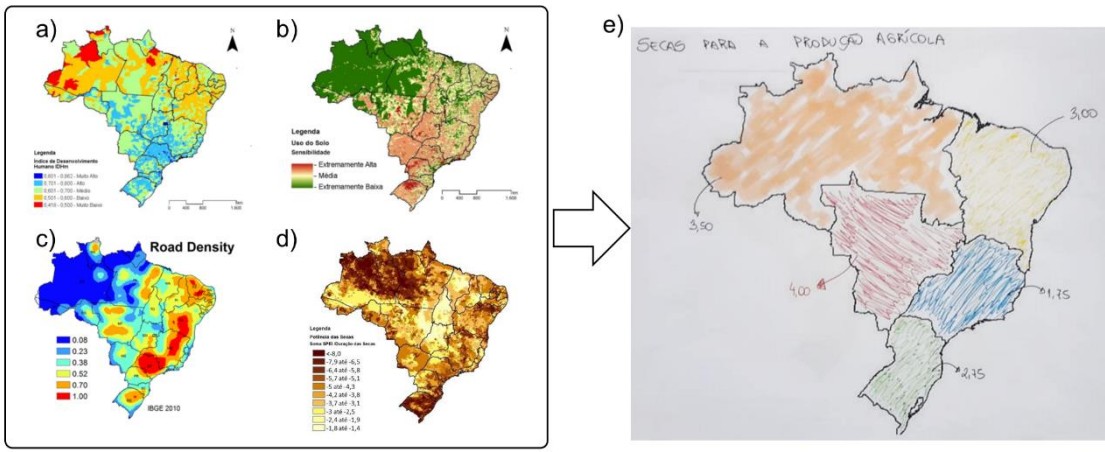


**Figure B8. Package of maps used in the second training for mapping climate risk associated with agricultural droughts. The maps illustrate: a) Human Development Index (HDI, MMA 2017); b) land use (MMA 2017); c) road density (Debortoli et al. 2017); d) drought potential (MMA 2017); and e) the climate risk map drawn by the participants.**




## Training 2

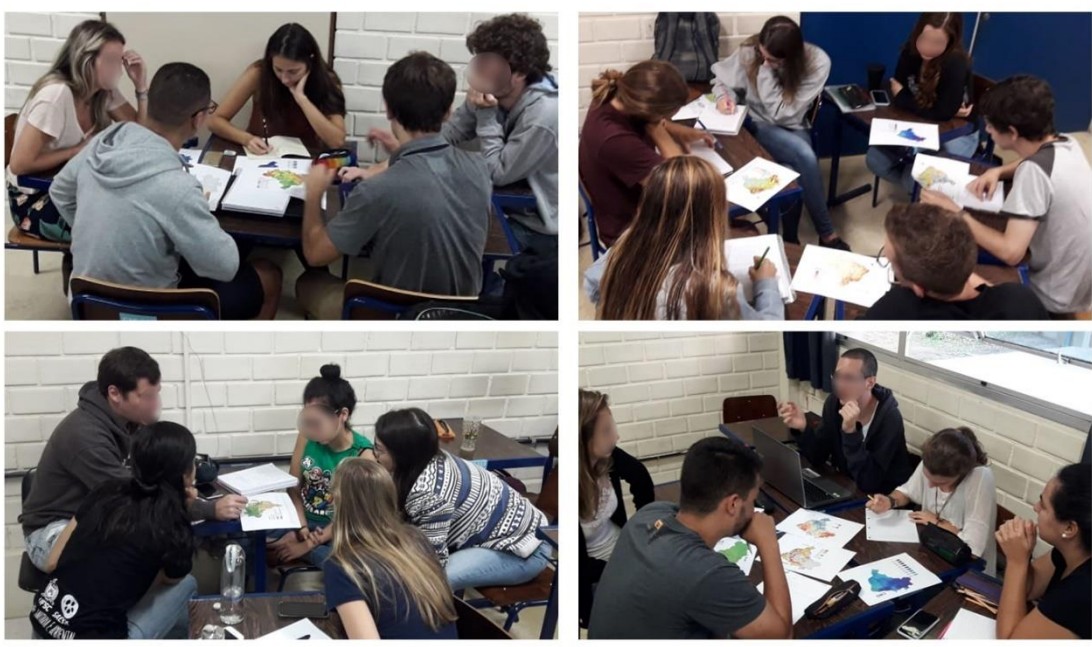

**Figure B9. Photos of the groups working on the activities of the second training.**

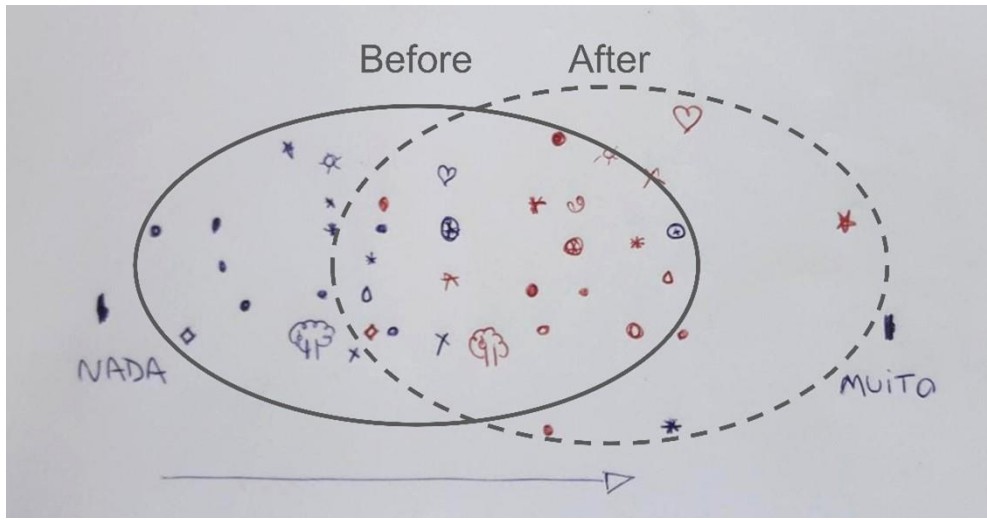

**Figure B10. Chart illustrating the evolution of participants' perception of learning about climate risk in the second training. Each symbol represents a participant and colours represent the temporality (before and after the lecture). The circle with continuous line compasses the perception of the participants at the beginning of the training, whereas the dashed line circle embraces the perception at the end of the lecture.**



**Training 3**

Date: July 5-6th 2019

Institution: State University of São Paulo (UNESP)

Number of participants: 19

Profile: graduate students from Water Resources Management Master programme

Duration: 11 hours

Teaching modules: introduction to climate change, introduction to climate information, climate risk analysis, and climate risk analysis (future)

Documentation available: maps, risk maps, photos of the activities, perception chart

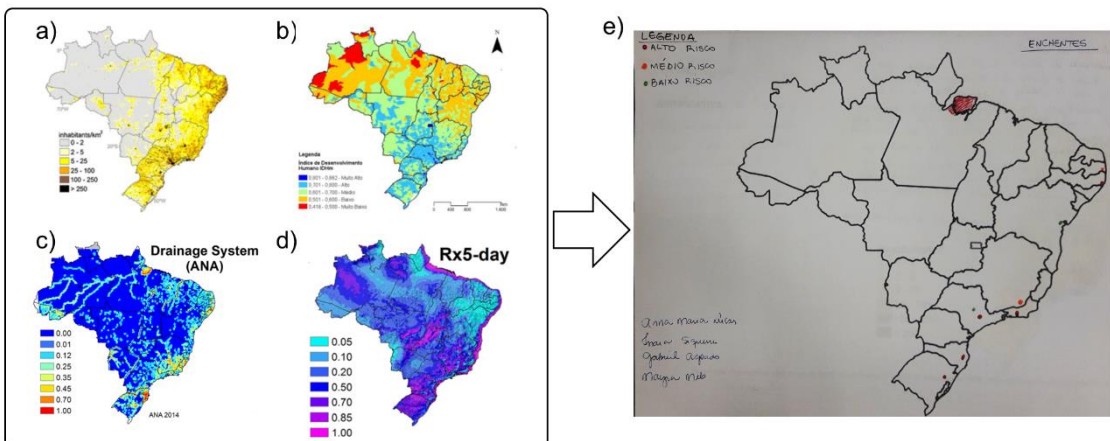

**Figure B11. Package of maps used in the third training for mapping climate risk associated with floods. The maps illustrate: a) Brazilian population density (Torres et al. 2012); b) Human Development Index (HDI, MMA 2017); c) watershed drainage (Debortoli et al. 2017); d) maximum consecutive 5-day precipitation (Rx5-day, Debortoli et al. 2017); and e) the climate risk map drawn by the participants**





## Training 3 | Working Group Landslides

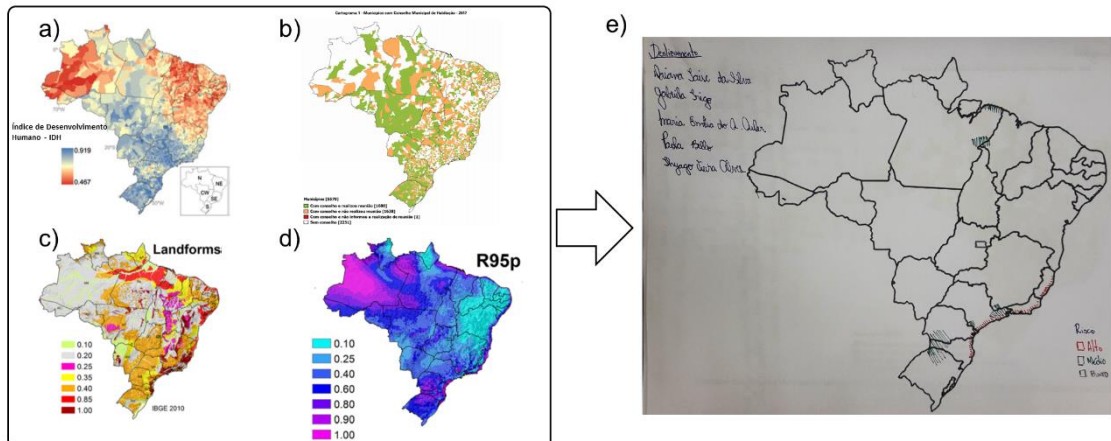


**Figure B12. Package of maps used in the third training for mapping climate risk associated with landslides. The maps illustrate: a) Human Development Index (HDI, Torres et al. 2012); b) municipal housing council (IBGE, 2017); c) landforms (Debortoli et al. 2017); d) annual total precipitation when rainfall > 95 percentile (R95p, Debortoli et al. 2017); and e) the climate risk map drawn by the participants.**


## Training 3 | Working Group Water Scarcity

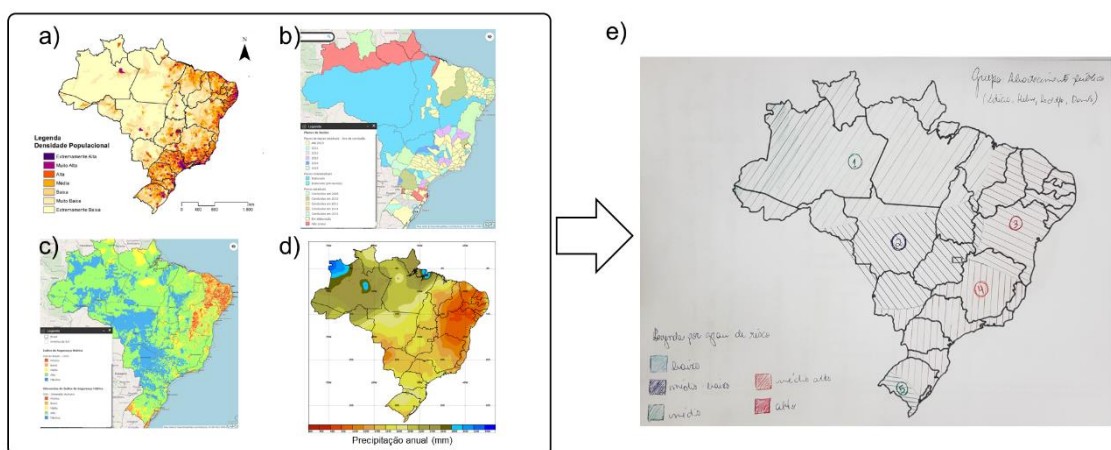

**Figure B13. Package of maps used in the third training for mapping climate risk associated with water scarcity. The maps illustrate: a) population density (MMA, 2017); b) stage of implementation of the water resources planning (Planos de Recursos Hídricos, 2020); c) water security index (Planos de Recursos Hídricos, 2020); d) annual rainfall (Normais climatológicas do Brasil,**
**2020); and e) the climate risk map drawn by the participants.**



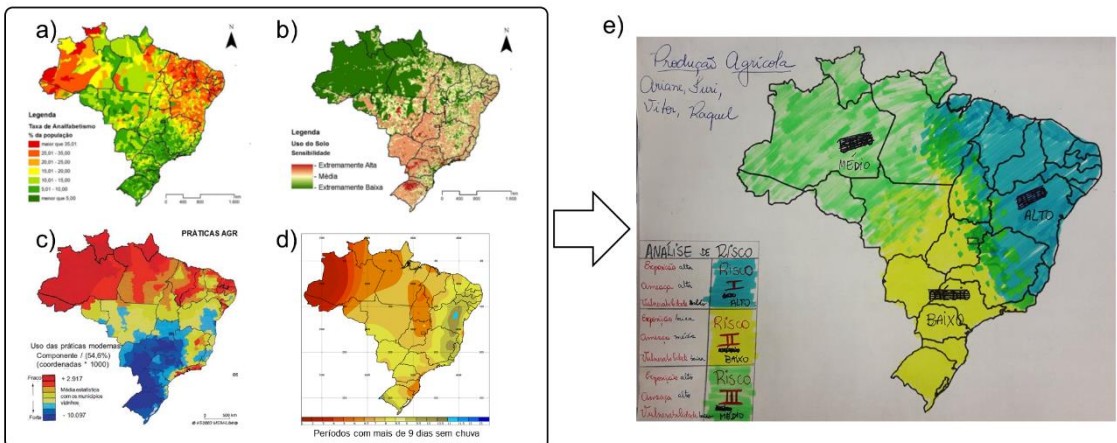

**Figure B14. Package of maps used in the third training for mapping climate risk associated with agricultural droughts. The maps illustrate: a) illiteracy rate (MMA 2017); b) land use (MMA 2017); c) use of modern practices in agriculture (Théry and Mello, 2018); d) number of dry spells longer than 9 days (Normais climatológicas do Brasil, 2020); and e) the climate risk map drawn by the participants.**


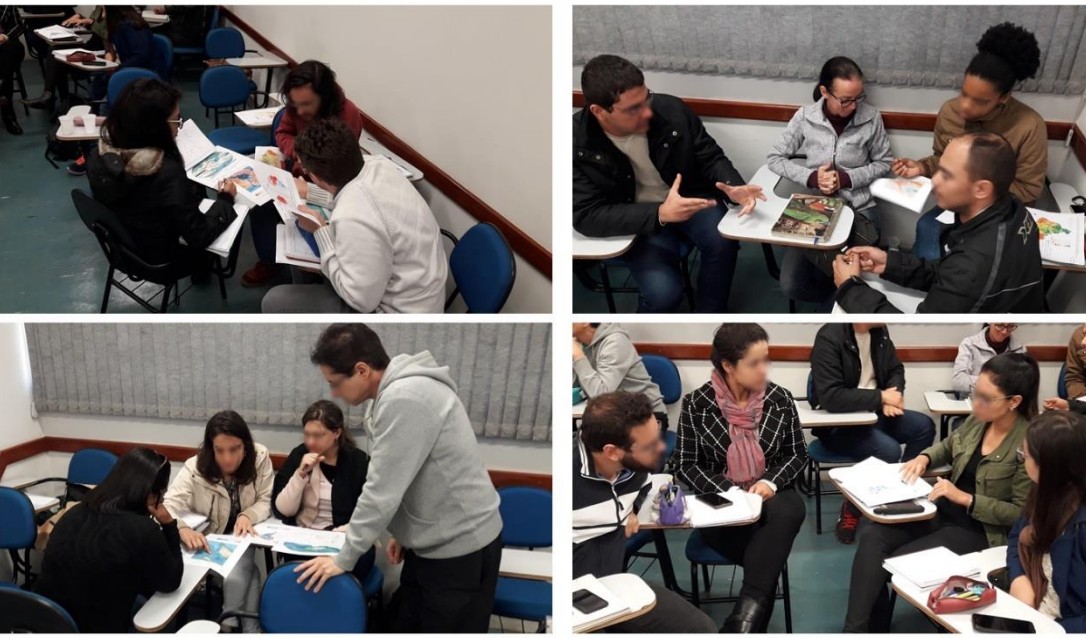

**Figure B15. Photos of the groups working on the activities of the third training.**






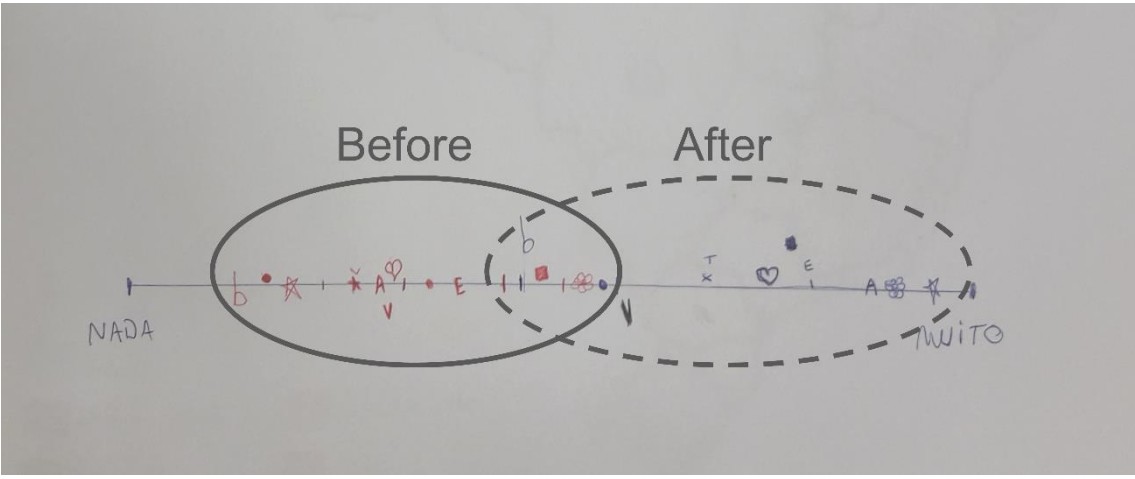

**Figure B16. Chart illustrating the evolution of participants' perception of learning about climate risk in the third training. Each symbol represents a participant and colours represent the temporality (before and after the lecture). The circle with continuous line compasses the perception of the participants at the beginning of the lecture, whereas the dashed line circle embraces the**
**perception at the end of the lecture.**

**Training 4**

Date: Aug. 29th, 2019

Institution: Federal University of Santa Catarina (UFSC)

Number of participants: 22

Profile: undergraduate students from Sanitary and Environmental Engineering

Duration: 2 hours

Teaching modules: Climate Risk Analysis

Documentation available: maps and photos of the activities





## Training 4 | Working Group Floods

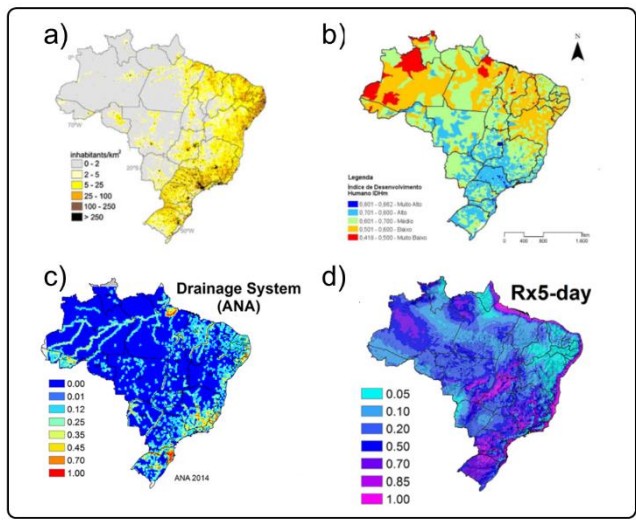


**Figure B17. Package of maps used in the fourth training for mapping climate risk associated with floods. The maps illustrate: a) Brazilian population density (Torres et al. 2012); b) Human Development Index (HDI, MMA 2017); c) watershed drainage (Debortoli et al. 2017); and d) maximum consecutive 5-day precipitation (Rx5-day, Debortoli et al. 2017)**

## Training 4 | Working Group Landslides

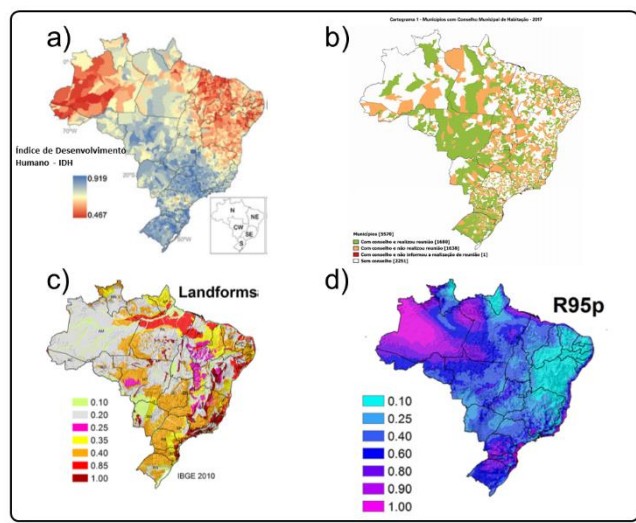


**Figure B18. Package of maps used in the fourth training for mapping climate risk associated with floods. The maps illustrate: The maps illustrate: a) Human Development Index (HDI, Torres et al. 2012); b) municipal housing council (IBGE, 2017); c) landforms (Debortoli et al. 2017); and d) annual total precipitation when rainfall > 95 percentile (R95p, Debortoli et al. 2017).**





## Training 4 | Working Group Water Scarcity

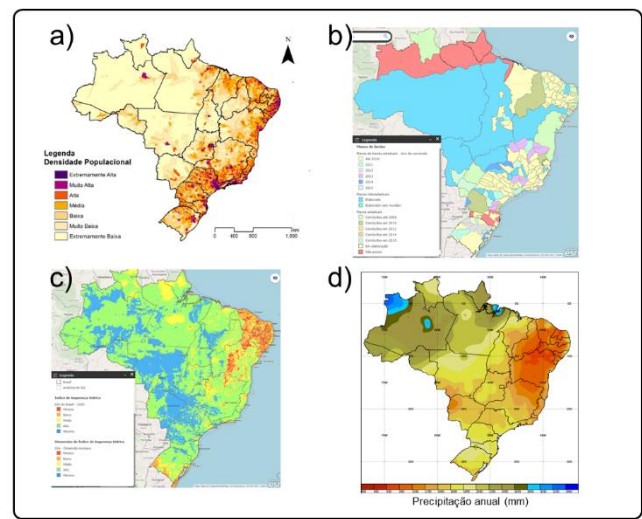


**Figure B19. Package of maps used in the fourth training for mapping climate risk associated with water scarcity. The maps illustrate: a) population density (MMA, 2017); b) stage of implementation of the water resources planning (Planos de Recursos Hídricos, 2020); c) water security index (Planos de Recursos Hídricos, 2020); and d) annual rainfall (Normais climatológicas do Brasil, 2020).**


## Training 4 | Working Group Agricultural Droughts

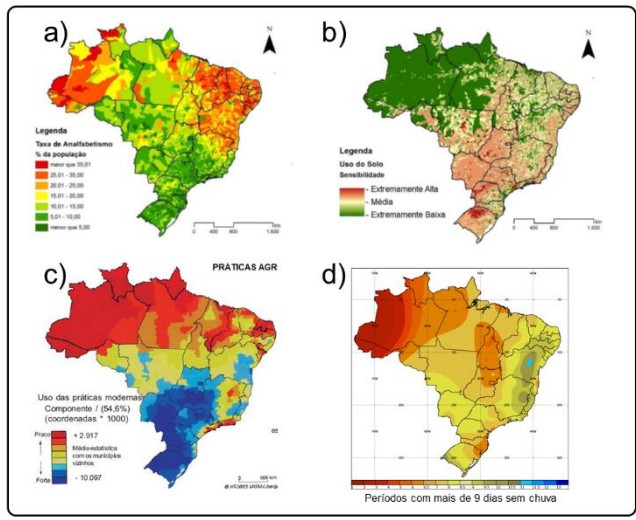

**Figure B20. Package of maps used in the fourth training for mapping climate risk associated with agricultural droughts. The maps illustrate: a) illiteracy rate (MMA 2017); b) land use (MMA 2017); c) use of modern practices in agriculture (Théry and Mello, 2018); and d) number of dry spells longer than 9 days (Normais climatológicas do Brasil, 2020).**




# Training 4

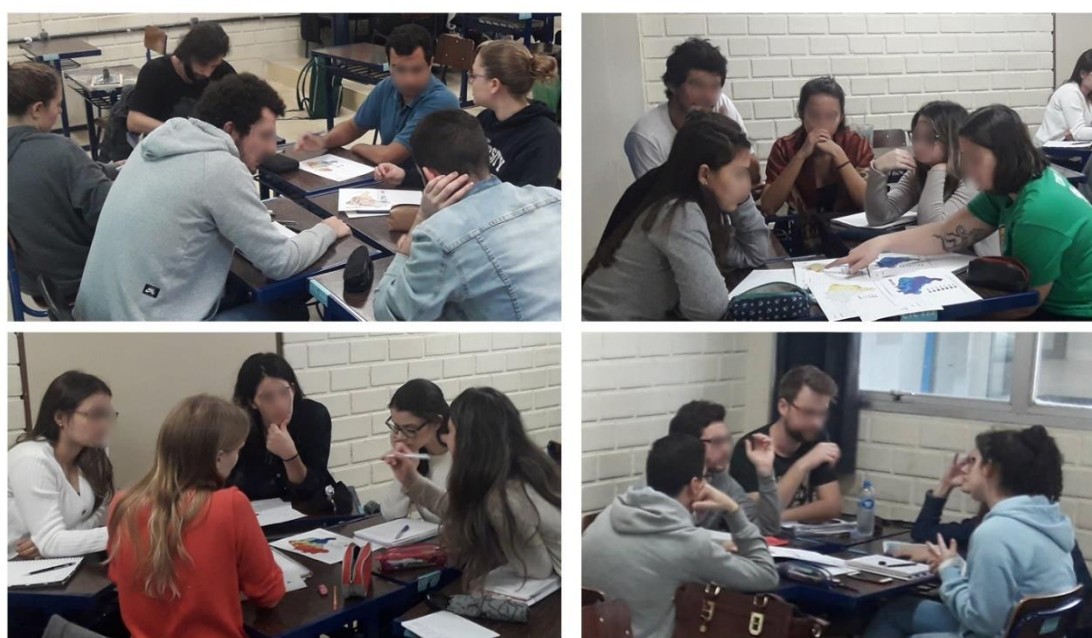

**Figure B21. Photos of the groups working on the activities of the fourth training.**

**Training 5**

Date: October 21st, 2019

Institution: Secretary of Sustainable Economic Development of Santa Catarina State (SDE)

Number of participants: 12

Profile: technicians and decision-makers

Duration: 4 hours

Teaching modules: Introduction to Climate Change and Climate Risk Analysis

Documentation available: maps and photos of the activities



## Training 5 | Working Group Water Scarcity

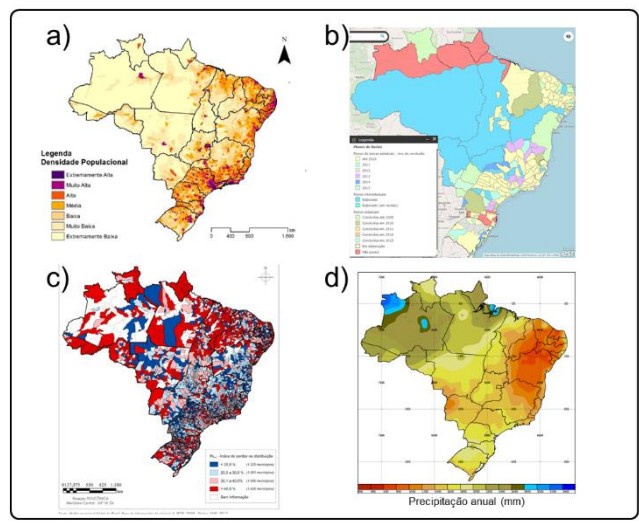

**Figure B22. Package of maps used in the fifth training for mapping climate risk associated with water scarcity. The maps illustrate: a) population density (MMA, 2017); b) stage of implementation of the water resources planning (Planos de Recursos Hídricos, 2020); c) water distribution loss index (Brasil, 2017); and d) annual rainfall (Normais climatológicas do Brasil, 2020).**

## Training 5 | Working Group Agricultural Droughts

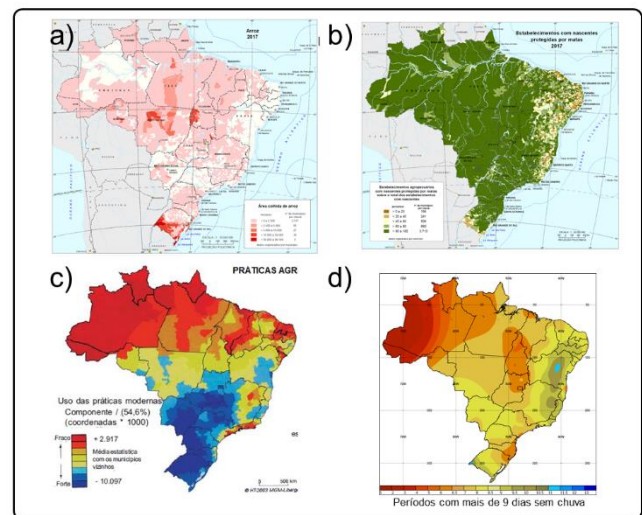

**Figure B23. Package of maps used in the fifth training for mapping climate risk associated with agricultural droughts. The maps illustrate: a) rice production (IBGE, 2019); b) properties with protected water springs (IBGE, 2019); c) use of modern practices in agriculture (Théry and Mello, 2018); and d) number of dry spells longer than 9 days (Normais climatológicas do Brasil, 2020).**





# Training 5

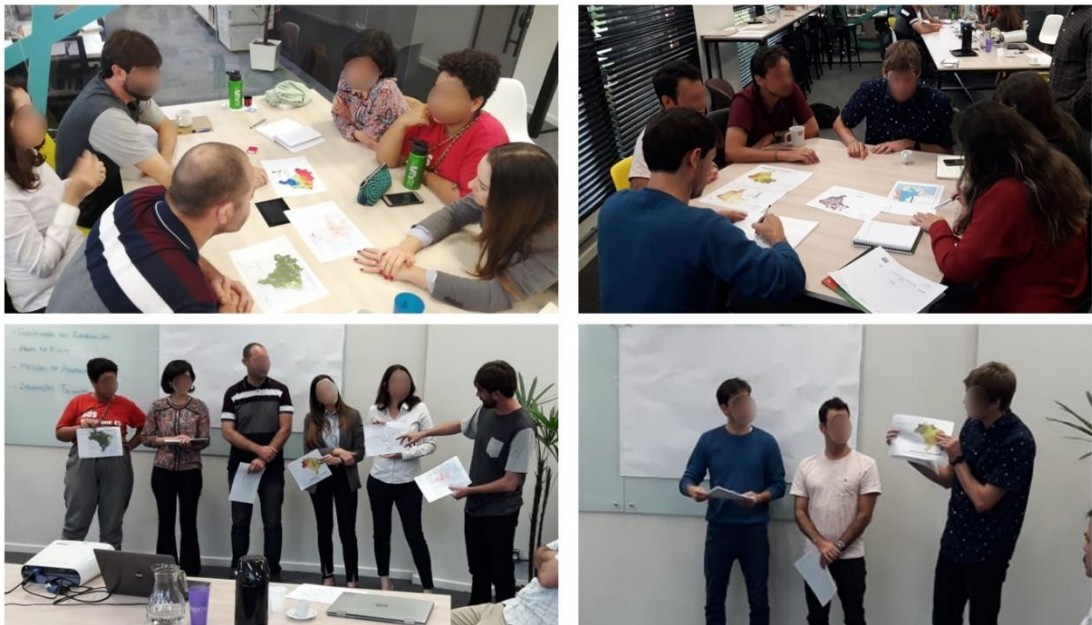

**Figure B24. Photos of the groups working on the activities of the fifth training.**