# Peer review of "Teaching climate risk for water planning: a pilot training for tertiary students and practitioners in Brazil"

_Geoscience Communication, 2021_

## Author Response (AR1)

**Author's response**

Pablo Borges de Amorim and Pedro Luiz Borges Chaffe

Author comment on "Teaching climate risk: a pilot training for tertiary students and

practitioners in Brazil" by Pablo Borges de Amorim and Pedro Luiz Borges Chaffe, Geosci.

Dear editor Dr. Sam Illingworth

We deeply appreciate the thoughtful and helpful review of our manuscript. We have adjusted the manuscript to account for all the referees' suggestions. The referees' constructive comments, suggestions, and corrections motivated us to further improve the manuscript.

We recognize that the current version of the manuscript is more coherent than the previous one and hope that in its current form the manuscript will be suitable for publication in the Geoscience Communication journal.

Please, consider our comments to each of the specific comments attached.

Yours sincerely,

Pablo Borges de Amorim and Pedro Luiz Borges Chaffe

**Reply to comments by Referee #1**

The authors appreciate the comments of Christopher Skinner (Referee #1) that helped us clarify and improve the main points of our manuscript. Please, find below our reply to all the comments.

Comment on gc-2021-23
Christopher Skinner (Referee)

Referee comment on "Teaching climate risk: a pilot training for tertiary students and practitioners in Brazil" by Pablo Borges de Amorim and Pedro Luiz Borges Chaffe, Geosci. Commun. Discuss., https://doi.org/10.5194/gc-2021-23-RC1, 2021

Thank you for the opportunity to review this manuscript. I found it to be extremely informative about an important topic- effectively communicating and building skills in climate change risks. It was also a pleasure to read.

I consider the manuscript to be relevant to the audience of *Geoscience Communication* and that it will be useful too. The manuscript covers the pedagogical development, delivery, and evaluation of a training course aimed at current and trainee water planners to help them process relevant climate information to assess risks. It details the pedagogical theory used throughout the workshops, namely problem based learning, and the qualitative observational methods used to evaluate them. Due to the nature of the journal, these ideas may be new to much of the audience but the authors have done a good job in explaining them clearly. Consequently, the manuscript serves as a useful introduction to these ideas to those seeking to develop similar training courses.

Overall, I have very few concerns about this manuscript and recommend it for publication after minor edits. The following are my suggestions for how the manuscript could be further improved.

The training workshops are focussed around water professionals (see line 58), either current practitioners or undergraduate students likely to pursue a career in water resource planning. The training was not tested on any groups outside of this area. It would be useful to the reader if this was made clear earlier in the manuscript, definitely in the introduction, ideally in the title and/or abstract. The introduction ought to

have a paragraph on why enhanced learning around climate risk and adaptation is particularly/specifically important for water professionals.

Reply: We appreciate the positive view on the manuscript. We have now made it clear that this training focuses on water professionals. We adjusted the title and the abstract, and we added a paragraph in the introduction about the importance of this kind of training for water professionals.

Lines 1-2 (Title): Teaching climate risk for water planning: a pilot training for tertiary students and practitioners in Brazil

Lines 12-15 (abstract): "In this case study, we present a short-duration face-to-face training for water professionals about the Intergovernmental Panel on Climate Change (IPCC)'s climate risk framework. The training uses Problem-Based Learning (PBL) pedagogy, and its suitability and benefits are evaluated with observational qualitative analysis and self-assessment of knowledge of the tertiary students and practitioners of 5 independent groups in Brazil."

Lines 47-55 (introduction): "Training tertiary students and practitioners on climate risk is useful for enhancing awareness about the relevance of the topic and informing about the existence of methods to reduce climate risks (George et al., 2009, 2016). It can help society plan and implement adaptation options with respect to the impacts of climate change (Fernandez et al., 2014), as foreseen in the SDG 13 (United Nations, 2016). Training in climate risk has been shown effective in developing knowledge, attitudes, and skills of farmers, meteorologists, managers, and policy makers around the world to make informed decisions to tackle climate change impacts (e.g., Yen et al., 2019; George et al., 2006, 2019). This is particularly important for water professional because the water sector is highly exposed to climate hazards and global warming is projected to further intensify the water cycle (Jiménez Cisneros et al., 2014). Especially in Brazil, where changes to a drier hydrological regime are expected in the future (Borges and Chaffe, 2019; Borges de Amorim et al., 2020)"

Similar to the point above, have the authors considered how applicable the training would be to groups outside of water professionals? A brief note in the discussion on this would help readers who may wish to use a similar training method to communicate climate risks to a different participants. For example, how important is it that the participants are water professionals? Could this be mitigated through additional teaching modules?

Reply: We have now added a paragraph in the discussion about the implications of delivering this training to the community outside the water resources field.

After line 302: "This training is designed for the water sector, but it can be easily adapted to other domains that require spatial planning and are exposed to climate hazards, such as health, agriculture, energy, transport, biodiversity, and ecosystems (Sherbinin et al., 2019; Nobre et al., 2019). The maps must be adjusted, and the trainers must ensure that all four elements that comprise climate risk (i.e., climate hazard, exposure, sensitivity, and adaptive capacity) are represented in the collection of maps. The interactive exercise in Session 1 can be used in other domains to help participants to assimilate the IPCC's climate risk concept. However, that might depend on the pre-existing knowledge and experience of the participants regarding floods (Mohadjer et al., 2021). Moreover, the classroom culture must be taken into consideration in this kind of training and adjustments may be necessary (Mohadjer et al., 2021; George et al., 2006)."

It may make the manuscript over length but I would be interested in some commentary about the differences, if any, in the responses between the groups. Did the undergraduates in Training 1 respond markedly different to the technicians and practitioners in Training 5?

Reply: In this case study, it was not possible to assess the differences in the self-assessment responses between all groups because the self-assessment was applied only for the undergraduate students from training 2 and 3. However, considering our observations and the feedback from the participants, it is unlikely that there were substantial differences in the responses between the groups. We have now adjusted the 'Methods of evaluation' and 'results' and 'discussion' sections to clarify this point.

After line 159: "The self-assessment was applied in training 2 and 3."
After line 174: "In general, it was observed that the participants from all groups could answer the questions correctly,…"

After line 213: "Figure 8 illustrates the students' self-rating in training 2 and 3."
After line 305: "In this case study, it was not possible to assess the differences, if any, in the self-assessment responses between the groups (e.g., undergraduates vs. practitioners) because the self-assessment was applied only for the undergraduate students (training 2 and 3). However, considering the classroom observations and the feedback from the participants, it is unlikely that there were substantial differences in the responses between the groups".
Figure 5: We adjusted the boxes of trainings 2 and 3. We replaced "Perception chart" to "Self-assessment".

Considering the audience of the journal it would be helpful for the authors to expand their introductions to some of the pedagogical theories. In particular: a brief description of active learning after line 34 and how it links into problem-based learning; a more developed introduction to problem-based learning (i.e., I'm not familiar with the literature around it but are there any criticisms of it?) after line 41; and, a more detailed description of the qualitative observation methods used after line 144.
Reply: We have now added a brief description of active learning and how it links into PBL after line 34: "Active learning is defined as any instructional method that promotes student activity and engagement in the learning process (Prince, 2004). An active learning method that is well suited to climate change education is Problem-Based Learning (PBL) (McCright et al., 2013)."

We added a more developed introduction to PBL after 41: "There is no consensus that students' scores are improved in PBL when compared to traditional learning and some professionals believe the advantages are negligible in comparison to the resources and preparation needed (Prince, 2004; Wood, 2003)."

We added a more detailed description of the qualitative observation methods after line 144: "Qualitative observation is a way to assess what students do (Grove et al., 2013). It provides rich and in-depth descriptions of classroom practices and are useful for identifying limitations of training schemes and educational courseware (Grove et al., 2013; Lindorff and Sammons, 2018). In this case study,…". We also adjusted the term "qualitative observational analysis" to "qualitative observation" through the manuscript.

Line 53-55: Helpful to add here that the training was face-to-face as this is not immediately obvious these days.
Reply: Indeed, it is an important point. We have now added that the training was "face-to-face"
Line 12: "In this case study, we present a short-duration face-to-face training for tertiary students and practitioners about the Intergovernmental Panel on Climate Change (IPCC)'s climate risk framework."
Lines 53-55: "In this paper, we describe the development and delivery of a short duration face-to-face training based on the IPCC's climate risk framework and PBL."
Line 311: "The training was delivered face-to-face to 5 independent groups in Brazil from 2018 to 2019 and reached 94 higher education students and practitioners in the field of water resources planning"

Lines 65-69: PBL has already been introduced so this section is repetitive. The extra information here could be included in the introduction.
Reply: We have adjusted the sentence and relocated the extra information to the introduction (Lines 36-39). To make clear that this training adopts PBL, we wrote in Line 67: "The theoretical basis underpinning this training is PBL and the background information used is the IPCC's climate risk framework."

Lines 105-130: It is unclear here whether the facilitator, or anyone else, is able to assist the work groups, either by providing advice or answering questions, or whether the groups were left to themselves and just observed. This would be useful to know when interpreting the results.
Reply: We have now noted that the trainer may assist the work groups by answering questions regarding the tasks and learning goals of the sessions, as well as additional information that is required for the tasks (e.g., interpretation of the maps).
Line 116: "The trainer assists the working groups in understanding the tasks and learning goals of the sessions. In this specific session, it is important to ensure that the working groups correctly interpret the maps (e.g., theme, legend, and caption) and understand how these data is associated with the IPCC's climate risk components.

Line 125: The authors state here that the groups were asked to classify the risk zones as low, medium, or high, yet later they state that earlier groups did not do this and only later groups did when told to do so. This should be clarified here.

Reply: We have now clarified that in the text.

Lines 124-125: "Each working group receives a blank map of Brazil where they are asked to illustrate risk zones. From the second training onwards, we informed about the possibility of using a scale of risk (e.g., low, medium, and high)."

We also added a note in the Session 5. "From the second training onwards, we provided instructions towards a more focused presentation."

Lines 177-179: Were the participants' queries answered during the sessions?

Reply: Yes, we have now clarified that in the text.

Lines 178-179: "The participants' queries were mainly about climate indices, such as the maximum number of consecutive dry days, and they were answered by the trainers during the session."

Line 223-228: This commentary on the efficacy of the self-assessment methods should be included in the discussion (Section 4.3?) rather than in the results section.

Reply: We have relocated this commentary to the discussion section (Lines 302-306).

Lines 249-252: How important is the facilitation script compared to the trainers experience the authors discuss a few sentences above? Are these two sections contradictory or are both elements required?

Reply: We have now adjusted the text to make it clear that the trainers' experience on the subject and facilitation skills are complementary and that both are necessary for the delivery of this training.

Line 248: "In addition to the trainers' experience on the subject, three pedagogical skills are necessary for the delivery of this training, which are: facilitation, ability to stimulate students, and preparation of educational courseware."

Lines 261-264: This is an important and timely comment but it was not immediately obvious (until you get to the Figures) that the training was face-to-face.

Reply: We have adjusted the text in order to make it clear that this training was face-to-face.

Line 261: "In addition, this training was delivered face-to-face, but trainers should consider specific pedagogical practices for online trainings"

Lines 311-313: Should include here that the students and practitioners were all in the field of water resource planning.

Reply: We have included that the students and practitioners were all in the field of water resource planning.

Lines 311-313: "The training was delivered to 5 independent groups in Brazil from 2018 to 2019 and reached 94 higher education students and practitioners in the field of water resources planning,…"

Figure 3: Add a note to description to say this is based on the IPCC climate risk framework.

Reply: We added a note to the caption that this is based on the IPCC climate risk framework.

Line 466: "The definitions of hazard, exposure, sensitivity, and adaptive capacity are based on the IPCC climate risk framework."

Figure 7: Some of the face blurring is not effective and participants could potentially be identified.

Reply: We have improved the face blurring of figures 7, A9, A15, A21 and A24.

Figure 8: I probably should say to state which colour denotes before and which after, but they swap between the training and the circles make it obvious, so I don't think it is required here.

Reply: We have excluded the sentence "and colors represent the temporality (before and after the training)".

**Reply to comments by Referee #2**

The authors appreciate the comments of Referee #2 that helped us clarify and improve the main points of our manuscript. Please, find below our reply to all the comments.

Comment on gc-2021-23
Anonymous Referee #2
Referee comment on "Teaching climate risk: a pilot training for tertiary students and practitioners in Brazil" by Pablo Borges de Amorim and Pedro Luiz Borges Chaffe, Geosci. Commun. Discuss., https://doi.org/10.5194/gc-2021-23-RC2, 2021

I like the manuscript. It is well written and well explained.
Reply: We appreciate the positive view of the manuscript.

My comments:
I'm wondering about the length of the article and I have seen some repetition in the manuscript text. It would be nice if authors remove sentences having the same meaning in different sections.
Reply: We have now thoroughly revised the text and removed the sentences having the same meaning in different sections.

In the introduction section, the authors describe more about climate risk and different approaches for teaching climate risk. However, adding some evidence for teaching climate risk (or any natural hazard and their risks) from around the globe could be useful. My concern here is: can the authors provide some examples that teaching climate, earthquake, flood, or any natural hazard to the students/public is helping to increase their awareness and to reduce the associated risk? I think there are some good examples for at least earthquake risk (e.g. https://doi.org/10.3389/feart.2020.00180, https://doi.org/10.5194/gc-4-281-2021). Why teaching climate risk is important, is it for reducing the risk or to motivate the public for the topic, or something else? I suggest adding a paragraph to explain teaching or training the public (students, any) is useful for…..
Reply: We have now added a paragraph in the introduction to explain why training on climate risk is important.
Lines 47-55: "Training tertiary students and practitioners on climate risk is useful for enhancing awareness about the relevance of the topic and informing about the existence of methods to reduce climate risks (George et al., 2009, 2016). It can help society plan and implement adaptation options with respect to the impacts of climate change (Fernandez et al., 2014), as foreseen in the SDG 13 (United Nations, 2016). Training in climate risk has been shown effective in developing knowledge, attitudes, and skills of farmers, meteorologists, managers, and policy makers around the world to make informed decisions to tackle climate change impacts (e.g., Yen et al., 2019; George et al., 2006, 2019). This is particularly important for water professional because the water sector is highly exposed to climate hazards and global warming is projected to further intensify the water cycle (Jiménez Cisneros et al., 2014). Especially in Brazil, where changes to a drier hydrological regime are expected in the future (Borges and Chaffe, 2019; Borges de Amorim et al., 2020)"

The authors mentioned the 'Environmental Engineering bachelor course of the Federal University of Santa Catarina (UFSC)' a couple of times but its brief description is missing. Furthermore, it would be nice if course content is added in the supplementary files.
Reply: We added a brief description of the course. If the referee does not mind, we included a reference to the course content instead of including in the supplementary files, because the course content is not our authorship.

Line 176: "The bachelor's course has a minimum duration of 10 semesters and includes basic subjects such as calculus, physics, and chemistry, as well as vocational subjects such as hydraulics, hydrology, and wastewater treatment. The course content is available at UFSC (2014)"

It would help readers to understand the scenario if authors discuss how students can play the role in water resource planning.
Reply: We have now added an extra topic in the discussion about how the students can apply the knowledge (acquired in this training) in water resource planning.

"**4.4. Application of the concept of climate risk by water professionals**
The concept of climate risk is useful for governmental bodies that are involved in long-term water resources planning and emergency preparedness, such as water agencies, ministries, water suppliers, civil defence and state executive offices (George et al., 2016; Flagg and Kirchhoff, 2018; Raucher et al., 2018; Yates et al., 2015; Boholm and Prutzer, 2017). In Brazil, examples of organizations are the National Water and Sanitation Agency (ANA and CGEE, 2014), the Ministry of Environment (Brazil, 2016), and the SDE (Santa Catarina, 2009). Universities, research centres, and consultancies are also interested in information about the impacts of climate change on the water resources (Borges de Amorim et al., 2020). Many water professionals are already adopting new analytic tools to respond to climate change, and the demand for climate risk training is increasing (Raucher et al., 2018; George et al., 2016; Yates et al., 2015). The challenge that remains in the application of the climate risk concept by water professionals is the lack of political saliency and unclear demarcation of responsibility between actors (Boholm and Prutzer, 2017; McBean and Rodgers, 2010). Training water professionals from governmental bodies, as we did with SDE, can raise awareness about the role of governmental bodies in climate change adaptation and, consequently, increase the demand for water professionals equipped with knowledge on climate risk (McNamara, 2013; Pruneau et al., 2013; McBean and Rodgers, 2010; George et al., 2016)."

A small paragraph on how such training would help the public (not from the related domain) would help readers to apply similar training ideas in different cultures/domains.
Reply: We have now added a paragraph in the discussion about the implications of delivering this training to the community outside the water resources field. We also cited the literature recommended above, i.e., Mohadjer et al. (2021).
After lines 301: "This training is designed for the water sector, but it can be easily adapted to other domains that require spatial planning and are exposed to climate hazards, such as health, agriculture, energy, transport, biodiversity, and ecosystems (Sherbinin et al., 2019; Nobre et al., 2019). The maps must be adjusted, and the trainers must ensure that all four elements that comprise climate risk (i.e., climate hazard, exposure, sensitivity, and adaptive capacity) are represented in the collection of maps. The interactive exercise in Session 1 can be used in other domains to help participants to assimilate the IPCC's climate risk concept. However, that might depend on the pre-existing knowledge and experience of the participants regarding floods (Mohadjer et al., 2021). Moreover, the classroom culture must be taken into consideration in this kind of training and adjustments may be necessary (Mohadjer et al., 2021; George et al., 2006)."

Line 50: The reason why authors chose Brazil is because of the high demand for climate risk experts…. While authors are talking about climate risk, it is better to provide some information about climate-induced natural hazards in Brazil.
Reply: We have now added information about climate-induced natural hazards in Brazil.
Lines 51-55: "We use Brazil as a case study as the country is already suffering considerable damages and losses associated with climate-induced natural hazards, particularly floods, landslides, droughts and heat waves (CEPED UFSC and World Bank, 2016; Nobre et al., 2019). At the same time, there is high demand for climate risk experts (Brazil, 2016) and a considerable number of courses and programs with the potential to include climate risk management (Cadastro Nacional de Cursos e Instituições de Educação Superior, 2020; Cursos da Pós-Graduação Stricto Sensu no Brasil [2017 a 2020], 2020)."

The climate risk definition is repeated. I suggest revising the text or remove the repeated part. (for example, line 70-75 and line ~100), same for PBL….
Reply: We have now thoroughly revised the text and removed the repeated parts.
The repeated explanation of PBL in the section 2.1 was removed and the extra information was relocated to the introduction (as asked by the Referee #1). We removed the sentence: "where risk is a combination of a climate hazard, with the exposure and vulnerability of a system. Vulnerability is comprised of sensitivity and adaptive capacity" (former lines 99-100)

With these minor modifications, the manuscript should be published.
Reply: We appreciate the important suggestions.